



# Improvement of Odin/SMR water vapour and temperature measurements and validation of the obtained data sets

Francesco Grieco[1], Kristell Pérot[1], Donal Murtagh[1], Patrick Eriksson[1], Bengt Rydberg[2], Michael Kiefer[3], Maya Garcia-Comas[4], Alyn Lambert[5], and Kaley A. Walker[6]

[1]Department of Space, Earth and Environment, Chalmers University of Technology, Gothenburg, Sweden
[2]Molflow, Gråbo, Sweden
[3]Karlsruhe Institute of Technology, Institute of Meteorology and Climate Research, Karlsruhe, Germany
[4]Instituto de Astrofísica de Andalucía, CSIC, Granada, Spain
[5]Jet Propulsion Laboratory, California Institute of Technology, Pasadena, CA, United States
[6]Department of Physics, University of Toronto, Toronto, ON, Canada

**Correspondence:** Francesco Grieco (francesco.grieco@chalmers.se)

**Abstract.**

Its long photochemical lifetime makes $H_2O$ a good tracer for mesospheric dynamics. Temperature is also an important tracer of seasonal circulation as well as multi-year trends. In this study we present the reprocessing of 18 years of mesospheric $H_2O$ and temperature measurements from the Sub-Millimetre Radiometer (SMR) on board the Odin satellite, resulting in a part of

the SMR version 3.0 level 2 data set. The previous version of the dataset showed poor accordance with measurements from other instruments, which suggested that the retrieved concentrations and temperature were subject to instrumental artifacts. Different hypotheses have been explored, and the idea of an underestimation of the single sideband leakage turned out to be the most reasonable one. The value of the lowest transmission achievable has therefore been raised to account for greater sideband leakage, and new retrievals have been performed with the new settings. The retrieved profiles extend between 40 - 100 km

altitude and cover the whole globe to reach 85° latitudes. A validation study has been carried out, revealing an overall better accordance with the compared instruments. In particular, relative differences in $H_2O$ concentration are always in the ±20% range between 40 and 70 km and diverge at higher altitudes, while temperature absolute differences are within ± 5 K between 40 - 80 km (with the exception of FM13 SMR–MLS difference reaching almost 10 K) and also diverge at higher altitudes.

## 1   Introduction

With a lifetime of the order of months at the stratopause and of a few days at 100 km, $H_2O$ is an important tracer of mesospheric circulation. It is also a main source of hydrogen radicals (such as OH, H, $HO_2$) that are involved in ozone destruction in the middle atmosphere (Brasseur and Solomon, 2005). A source of $H_2O$ in the mesosphere is methane oxidation:

$$CH_4 + 2 \cdot O_2 \rightarrow 2 \cdot H_2O + CO_2 \tag{1}$$

which, mostly important in the stratosphere, declines with altitude due to smaller abundances of methane and stops between

60 - 70 km. The assumption that one $CH_4$ molecule leads to two $H_2O$ molecules is likely not to be valid at altitudes above



60 km, where it would lead to an overestimation of $H_2O$ production (Frank et al., 2018). Another major source of $H_2O$ in the stratosphere is the uplift of moist air in the tropical tropopause. Due to vertical transport, this moist air can also reach the mesosphere and affect local $H_2O$ abundance. Above 70 km, another noteworthy source is oxidation of molecular hydrogen through two reactions:

$$H_2 + O \rightarrow H_2O \tag{2}$$

$$H_2 + O_3 \rightarrow H_2O + O_2. \tag{3}$$

The only major sink of $H_2O$ in the mesosphere is photodissociation:

$$H_2O + h \cdot \nu \rightarrow H + OH \quad (\lambda < 200\ \text{nm}) \tag{4}$$

which becomes more important with altitude and dominates over production above 70 km, resulting in a decrease of $H_2O$ concentration with increasing altitude. Moreover, the effect of vertical motions due to the meridional circulation play a major role. The downwelling during polar winter and the upwelling during polar summer result, respectively, in lower $H_2O$ concentrations in the upper stratosphere and mesosphere during polar winters and in larger $H_2O$ concentrations during polar summers (Lossow et al., 2019). Moreover, the very cold polar summer mesopause is favourable for the formation of polar mesospheric clouds (Pérot et al., 2010; Christensen et al., 2016). The deposition of water around 85-90km to form these clouds leads to a decrease of water vapor at those altitudes. The ice particles grow, sediment, reach the warmer regions at lower altitudes, where they sublimate, leading to an increase of water vapor around 80km.

Satellite observations of $H_2O$ in the middle atmosphere have been performed since the 1970s with the launch of Nimbus-7 satellite and the activity of two instruments on board: LIMS (Limb Infrared Monitor of the Stratosphere) (Remsberg et al., 1984) and SAMS (Stratospheric and Mesospheric Sounder) (Munro and Rodgers, 1994). Various instruments have followed through the years, the most recent ones being SABER (Sounding of the Atmosphere using Broadband Emission Radiometry) (Feofilov et al., 2009; Rong et al., 2019) launched on board TIMED (Thermosphere-Ionosphere-Mesosphere Energetics and Dynamics) in 2001, ACE-FTS (Atmospheric Chemistry Experiment - Fourier Transform Spectrometer) (Nassar et al., 2005) and MAESTRO (Measurement of Aerosol Extinction in the Stratosphere and Troposphere Retrieved by Occultation) (Sioris et al., 2010) launched on board Scisat-1 in 2003, and MLS (Microwave Limb Sounder) (Waters et al., 2006) launched on board the Aura satellite in 2004. All of these instruments are still operating. Moreover, in 2002 three instruments performing middle atmospheric $H_2O$ observations were launched on board the Envisat satellite: GOMOS (Global Ozone Monitoring by Occultation of Stars) (Montoux et al., 2009), MIPAS (Michelson Interferometer for Passive Atmospheric Sounding (e.g., Fischer et al., 2008) and SCIAMACHY (Scanning Imaging Absorption Spectrometer for Atmospheric Chartography) (e.g., Weigel et al., 2016). Their activity stopped in April 2012 due to loss of contact with the satellite.

The Sub-Millimetre Radiometer (SMR) on board the Odin satellite has been performing $H_2O$ measurements in the middle atmosphere since its launch in 2001 and is still operating. Previous studies using SMR $H_2O$ observations have been carried out





by Lossow et al. (2007, 2008, 2009) and Urban et al. (2007). These studies refer to SMR v2.1 L2 data retrieved from the 556.9 GHz $H_2O$ line. These profiles, in the mesosphere, present high biases compared to other instruments, i.e. around 20% between 40 - 70 km and greater than 50% between 70 - 100 km (Murtagh et al., 2020).

Together with $H_2O$, temperature is also a retrieval product obtained from the same spectra and it represents another useful tool

to study the middle atmospheric dynamics. Moreover, it is known that the global temperature is affected by the increase in greenhouse gases, not only in the troposphere, but also at higher altitudes. This has raised interest for investigating long-term temperature trends in the middle atmosphere. The SMR v2.1 temperature retrieved together with $H_2O$ shows biases up to 15 K in the mesosphere (Murtagh et al., 2020). One of the first studies of this kind based on satellite observations was carried out using data from SME (Solar Mesosphere Explorer) (Clancy and Rusch, 1989). Apart from SMR, in recent years other

satellite instruments have been employed for temperature measurements of this atmospheric region. Among these there are MLS (Microwave Limb Sounder) (Azeem et al., 2001) and HALOE (Halogen Occultation Experiment) (Hervig et al., 1996) launched in 1991 on board UARS (Upper Atmosphere Research Satellite), TIMED/SABER (Cooper, 2004), Envisat/MIPAS (Kiefer et al., 2020), Aura/MLS (Schwartz et al., 2008), Scisat-1/ACE-FTS (Sica et al., 2008) and SOFIE (Solar Occultation for Ice) (Liu et al., 2014) launched in 2007 on board satellite AIM (Aeronomy of Ice in the Mesosphere).

The Odin/SMR data set has undergone a full reprocessing, leading to a new version (v3.0). The present study, carried out to identify the instrumental origins of the above-mentioned biases, is part of this extensive reprocessing work. The improved retrieval method will be described in Section 2. The resulting $H_2O$ and temperature data sets are presented in Section 3 and validated in Section 4 by comparing them with independent satellite measurements from MIPAS, ACE-FTS and MLS.

## 2 Odin/SMR $H_2O$ and temperature measurements

### 2.1 The sub-mm radiometer

The Sub-Millimeter Radiometer (SMR) is an instrument on board the Odin satellite performing limb sounding of the middle atmosphere. The measurements cover the whole globe including the polar regions. Odin was launched on 20 February 2001 as a Swedish-led project in collaboration with Canada, France and Finland. Its 600 km sun-synchronous orbit has an inclination of 97.77° and a 18:00 hrs ascending node (slightly varying with time). SMR has four sub-millimeter receivers covering fre-

quencies between 486 - 504 GHz and 541 - 581 GHz and a millimeter receiver measuring radiation around 118 GHz, so that emissions from $O_3$, $H_2O$, CO, NO, ClO, $N_2O$, $HNO_3$ and $O_2$ due to rotational transitions can be detected (Frisk et al., 2003). SMR components are schematised in Figure 1. A Dicke switch allows to rapidly change the source of input radiation between the main beam and calibrators (cold sky and hot load); the radiation is then split according to polarisation and collected by different receivers where it is combined with a local oscillator (LO) signal by means of a mixer, converting the signal to lower

frequencies (3.3 - 4.5 GHz) and maintaining only the contribution from two sidebands. SMR is a single sideband instrument; it uses a Martin-Puplett interferometer with arm lengths tuned to optimize transmission of the primary band (containing the





signals of scientific interest) while suppressing transmission of the image band. The response of the interferometer with regards to frequency $\nu$ is equal to:

$$r = r_0 + \frac{(1 - 2r_0)}{2} \left[ 1 + cos\left( \frac{4\pi l\nu}{c} \right) \right],$$ (5)

where $l$ is the interferometer length, and $r_0$ is the lowest transmission value achieved (Eriksson and Urban, 2006) which is

not zero because it is not possible to achieve perfect suppression. The linear dependency of $l$ with respect to the temperature on board the satellite is also taken into account:

$$l(T) = l_0 + \frac{l_{sb}}{2} + c_T(T - T_0)$$ (6)

where T is the temperature of the satellite, $l_0$ is the interferometer length at the reference temperature $T_0$, $l_{sb}$ the nominal sideband path tuning length (expressed for both arms altogether, hence the division by 2), and $c_T$ is the coefficient of thermal

expansion. $l_0$, $T_0$ and $c_T$ values have been estimated by Eriksson and Urban (2006) from fits based on various observations. Since it is impossible to completely suppress the image band contribution, a sideband leakage ($p$) is included in the measurement, where $p$ is defined as:

$$p(\nu) = \frac{r(\nu')}{r(\nu) + r(\nu')}$$ (7)

with $\nu$ and $\nu'$ being, respectively, the primary band and image band center frequencies. Eventually, the signal is amplified

and directed to the spectrometers.

The observation time of the instrument has been equally shared between astronomical and atmospheric observations until 2007. After that, the astronomical mission was concluded and the instrument has been exclusively employed to perform atmospheric measurements. SMR measures spectra during upward and downward vertical scanning, allowing to scan the limb of the atmosphere from the upper troposphere to the lower thermosphere. However, in this study, we consider only mesospheric

measurements ranging from 40 to 100 km.

### 2.2    SMR $H_2O$ and temperature measurements: description and improvement

SMR receivers can be set up to cover different frequency bands. To each of these configurations, called frequency modes (FM), are assigned scheduled observation times. In this study we focus on mesospheric observations of the 556.9 GHz emission line from the corresponding $H_2O$ rotational transition. They are performed with a 3 - 4 km vertical resolution, using FMs 13 and

19 (while stratospheric observations are performed using other FMs) whose characteristics are summarised in Table 1. With these FMs, temperature and $O_3$ are retrieved although the latter is not the focus of this paper. The retrieval of temperature is made possible by the fact that the 556.9 GHz $H_2O$ emission line is saturated up to around 90 km (Murtagh et al., 2020).

The two FMs use different frontends, that is the set of components denoted by B2 and A1 in Figure 1. The retrievals were carried out using the Atmospheric Radiative Transfer Simulator (ARTS) which is a software package for long wavelenght


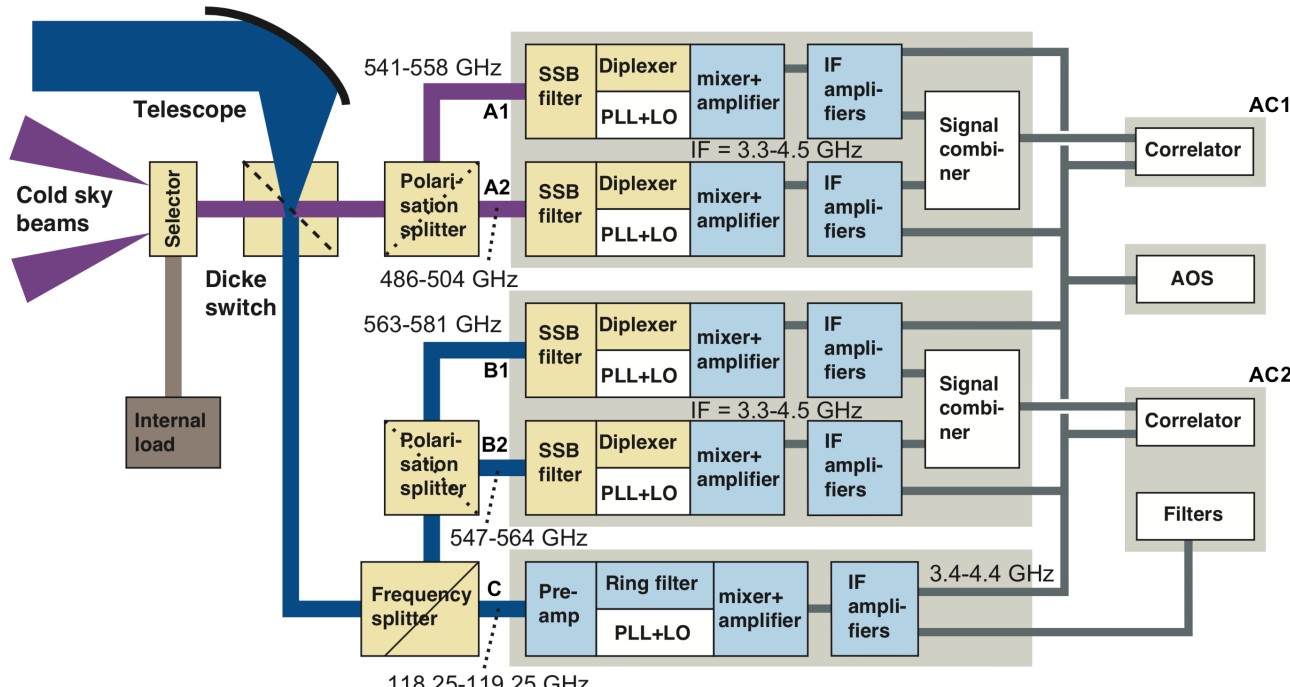

**Figure 1.** Block diagram of the Odin radiometer. From Frisk et al. (2003).

**Table 1.** Characteristics of the frequency modes (FM) used to observe mesospheric $H_2O$ and temperature. From Rydberg et al. (2017).

| Spectrometer | Frontend | LO Freq. [GHz] | Freq. Range [GHz] | Product | FM |
|:---:|:---:|:---:|:---:|:---:|:---:|
| AC1 | 555 B2 | 553.298 | 556.598 - 557.398 | $H_2O, O_3, T$ | **13** |
| | 549 A1 | 553.050 | 556.550 - 557.350 | $H_2O, O_3, T$ | **19** |

radiative transfer simulations, with a focus on passive microwave observations, incorporating effects of sensor characteristics (e.g. Eriksson et al., 2011). ARTS retrieval algorithms are based on the Optimal Estimation Method (e.g. Rodgers, 2000). In this study, a mesospheric inversion mode was used, performing retrievals from measurements with tangent altitudes between 40 and 100 km. A more detailed description of the retrieval process for v3.0 can be found in (Grieco et al., 2020). The temperature a priori is given by the ERA-interim reanalysis data (Dee et al., 2011) up to 60 km, while the Mass Spectrometer Incoherent Scatter model (version NRLMSISE-00; Picone et al., 2002) is used above 70 km. Between 60 - 70 km, a spline interpolation of the two is applied. The a priori for water vapour is a compilation from the Bordeaux Observatory. Both temperature and water vapour a priori are made available together with the retrieved profiles.





In the previous data version, $H_2O$ profiles retrieved from FMs 13 and 19 showed significant differences from measurements from other satellites. FM19 presented a bias between $\pm 20\%$ between 40 and 80 km, while FM13 presented concentrations around 10% higher than Scisat-1/ACE-FTS and Aura/MLS between 40 and 60 km, and around 20% higher than Envisat/MIPAS in the same altitude range. Both FMs showed differences greater than -50% between 80 and 100 km. Temperature biases were

also observed: in particular, for FM 19, the bias was equal to about -5 – -10 K between 60 and 80 km; while, for FM 13, the bias was equal to about +10 K between 40 and 80 km. Both FMs were characterised by very high negative biases at high altitudes. These differences can be seen in Figures 12 and 13, as well as in Murtagh et al. (2020) who used a smaller data set for comparison. This suggested the presence of instrumental artifacts. We investigated for possible non linearity in the spectra and for erroneous estimations of the pointing offset of the instrument, however an underestimation of $r_0$ (see Eq. 5) turned out

to be the most likely cause of the incongruous retrieved quantities. Sideband leakages greater than the nominal value have been already observed in spectra in Eriksson and Urban (2006). A $r_0$ of -14 dB had previously been assumed for both frontends used in the two FMs under consideration: an underestimation that caused spurious signal originated from the sideband leakage to be considered as part of the signal of interest, leading to misestimation of retrieved concentration and temperature. Setting the $r_0$ value to -13 dB for FM 13 and to -11 dB for FM 19 gave the best results in terms of minimizing the differences with

measurements from other instruments (see Section 4).

$H_2O$ retrieval for an exemplary scan is shown in Figure 2. A measure of how much a retrieved quantity is influenced by the a priori is given by the measurement response, a quantity defined as the sum over the row of the averaging kernel matrix (Rodgers, 2000). Data with a measurement response lower than 0.75 are discarded. This is the case for the retrieved profile above 100 km which is, nevertheless, shown here out of completeness as well as high altitudes averaging kernels.

## 3 The new data sets

In this section we present the new $H_2O$ and temperature products retrieved from FMs 13 and 19 measurements, which are part of the SMR v3.0 L2 data set. These data sets have been obtained using the improved retrieval algorithms, as described in the previous section. In particular we describe $H_2O$ concentration and temperature time series and compare the retrieved profiles to the previous v2.1 data set.

Figure 3 shows a histogram summarising the number of L1 and L2 products available for FM13 and FM19 during the whole Odin operational time period. The amount of L2 data is generally lower than L1 data since the retrieval process does not succeed for every scan. While from 2006 the two FMs have been used in similar proportion, in the earlier years, FM13 was used only occasionally, with a particularly high number of measurements performed during July 2002, July 2003 and August 2004. These are associated to a special scheduling set to study dynamics in the northern summer mesosphere related to the

presence of noctilucent clouds (Karlsson et al., 2004).

Figures 4 and A1 show time series of $H_2O$ volume mixing ratios corresponding to FM13 and FM19, respectively, in form of monthly zonal means over 5 latitude bands covering the whole globe. The equivalent figures for retrieved temperature are shown in figures 5 and A2. The gaps in the data sets observed globally every northern summer from 2013 are due to the





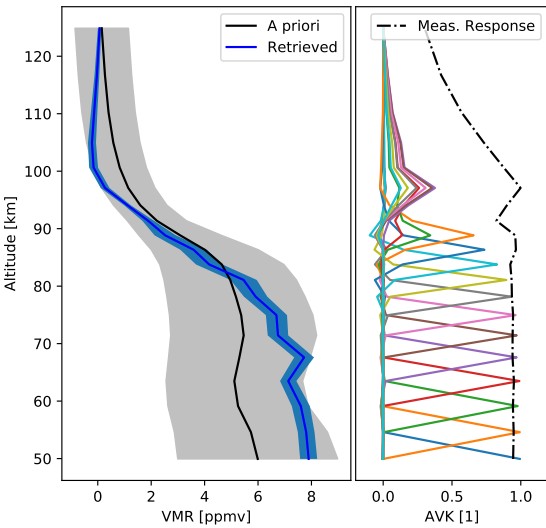

**Figure 2.** Example retrieval referring to ScanID 230753751 from FM 19. Left: Retrieved concentration profile and error due measurement thermal noise (Rodgers, 2000) (in blue and shaded, respectively) and a priori including uncertainties (in black and shaded, respectively). Right: Averaging kernels plotted in a different colour for each altitude (not indicated) and measurement response (dashed-dotted black line).

fact that the aging instrument is put in standby mode during the eclipse season, in order to save the batteries. In the tropics, we observe a clear Semi-Annual Oscillation (SAO) pattern above 75 km, in both the $H_2O$ and temperature data sets. This phenomenon is caused by SAO of the zonal winds in the mesosphere which is driven by momentum deposition from gravity and Kelvin waves coming from lower altitudes. Zonal winds SAO in turn give rise to SAO in meridional and vertical advection

(Hamilton, 2015). In particular, in correspondence of equinoxes, a sinking or weaker rising motion of air in the tropics causes the presence of lower $H_2O$ concentrations in the mesosphere; while stronger rising motion, occurring during solstices, cause higher $H_2O$ concentrations (e.g., Lossow et al., 2017). High latitudes are instead dominated by an annual cycle that features, at all altitudes and for both hemispheres, higher concentrations during local summertime, caused by upward transportation of moist air and increased methane oxidation due to the greater amount of received solar radiation; and lower concentrations

during local wintertime, due to the descent of dry air from the upper mesosphere via the downward branch of the mesospheric residual circulation. The amplitude of this oscillation is bigger in the southern hemisphere, where the descent of air is stronger and more stable (Schoeberl and Newman, 2015). Temperature maxima are observed in summer in the stratopause region while, in the upper mesosphere, the summer seasons correspond to temperature minima. This is explained by the fact that important gravity wave forcing pushes away this high altitude region from geostrophic balance, leading to the mesospheric residual

circulation. This circulation pattern is associated with upward motion of air during summertime, as already mentioned earlier, which results in a strong adiabatic cooling (Brasseur and Solomon, 2005). Moreover, in the northern high latitudes, some



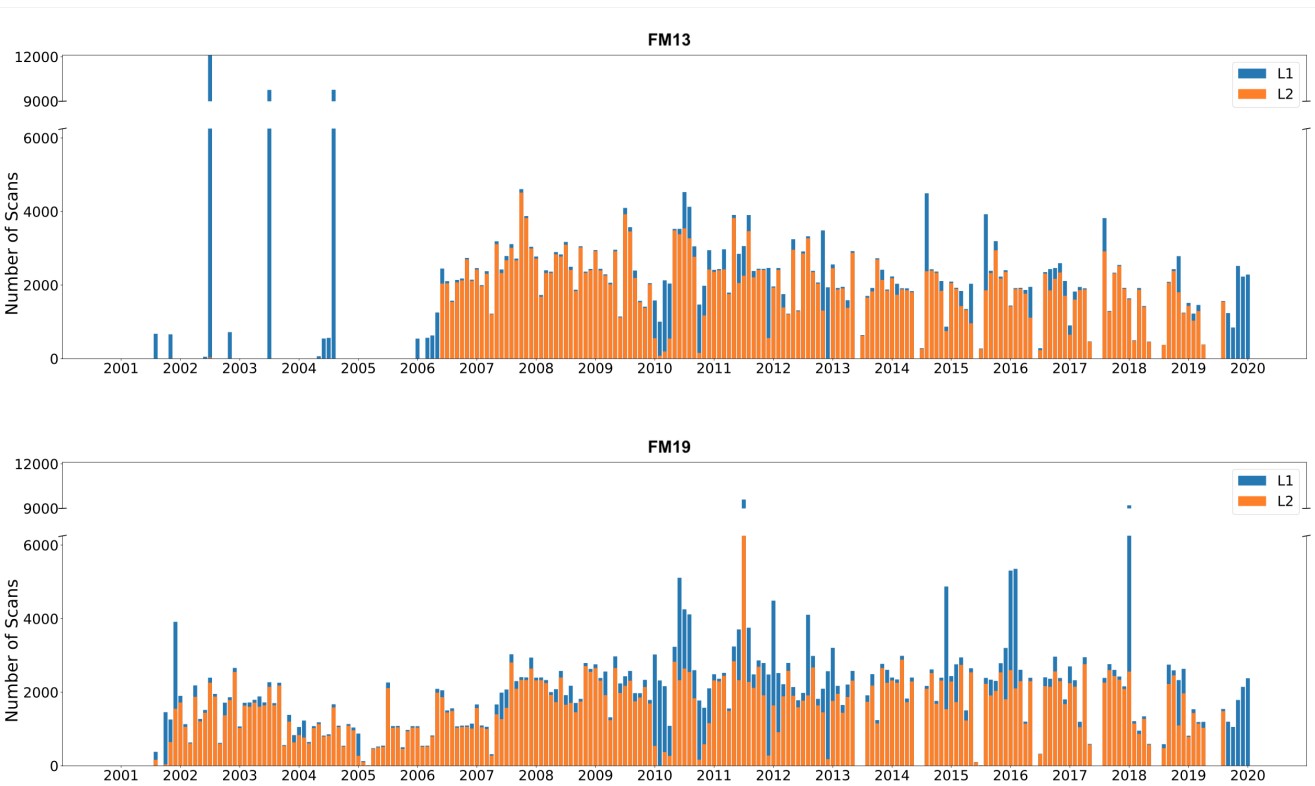

**Figure 3.** Number of L1 and L2 scans by month for FM13 (top) and FM19 (bottom). The ticks on the x-axis correspond to January 1st for each year.

particular features are observed in some years (namely 2004, 2006, 2009, 2013 and 2019), during late winter. Each is due to a Sudden Stratospheric Warming (SSW) followed by the formation of an elevated stratopause (Vignon and Mitchell, 2015). We can first observe secondary maxima in $H_2O$ concentration, corresponding to the onset of the SSW events, during which the polar vortex was disturbed by planetary waves (e.g. Charlton and Polvani, 2007). Following these SSWs, the polar vortex

5    recovered and the stratopause reformed at higher altitudes than normal, corresponding to the peaks in temperature observed around 80 km in Figure 5 and A2. Such events are associated with increased downward motion of air in the mesosphere (e.g. Pérot et al., 2014). This leads to transport of dry air from the upper mesosphere down to the lower mesosphere, as seen in Fig. 4 and A1. Mid latitudes show, in a less pronounced way, both effects of SAO and annual cycle. Finally, at all latitudes, lower $H_2O$ concentrations can be observed at high altitudes during the period 2012-2016 due to increased photolysis related to

10   a stronger solar activity. On the other hand, at low altitudes, higher $H_2O$ concentrations can be observed due to increased $O_2$ photolysis and consequent enhanced $CH_4$ oxidation (e.g., Remsberg et al., 2018).





**Figure 4.** Time series of FM19 $H_2O$ volume mixing ratios measured by SMR for different latitude bands. The white areas indicate periods and altitudes at which the number of measurements in the given latitude band is lower than 10. The ticks on the x-axis correspond to the beginning of each year.





**Figure 5.** Time series of FM19 temperature measured by SMR for different latitude bands. The white areas indicate periods and altitudes at which the number of measurements in the given latitude band is lower than 10. The ticks on the x-axis correspond to the beginning of each year.





In Figures A3 and A4, we compare the data sets corresponding to the two frequency modes considered in the study, for $H_2O$ and temperature, respectively. As explained in Section 2.2, those correspond to measurements made using different parts of the instrument. They should therefore be treated as two different data sets. Here, non-coincident profiles are compared, since the temporal and geographical coverage is different for the two frequency modes. Differences are therefore to be expected. These

comparisons are simply shown with the aim of summarising the average differences between both FMs, which could be useful information for the future users of these data sets. In Figure A3, the v3.0 FM13–FM19 $H_2O$ relative difference is shown. It is equal to -15% at 40 km altitude; the value then reaches 0% at 50 km and remains approximately constant until 60 km. It then increases to reach +100% at 85 km altitude and finally decreases back to 0% at 100 km altitude. The high relative difference values around 85 km are observed at all latitudes and seasons (not shown). Moreover, the v3.0 FM13–FM19 absolute differ-

ence with regards to temperature (Figure A4) oscillates between $\pm 8$ K and only reaches -16 K around 100 km. The relative difference values are referred to the mean between the FM13 and FM19 products.

Comparison of SMR v3.0 with respect to the older SMR v2.1 $H_2O$ dataset (e.g., Urban et al., 2007) shows, for FM13, a relative difference of -25% around 40 km which goes down to 0% at 60 km. This corrects for the v2.1 FM13 bias shown with

respect to ACE and MLS (see Section 2.2).The value stays around 0% until 80 km and then increases up to +20% at 90 km, to finally decrease to -7% at 100 km (Figure 6a). The v3.0 FM13 higher concentrations above 80 km result in a decrease of the high negative bias that characterised v2.1, although differences with respect to other instruments remain high at these altitudes, as discussed in Section 4. Zonal mean plots of difference values averaged over the whole SMR operating time period, for different latitudes and seasons, are shown in the Appendix. Peaks of +30% are registered during northern spring and summer,

as well as during southern summer and during local autumn in both hemispheres, at high latitudes between 80 - 100 km. Moreover, the highest negative values, of -60%, are observed during local autumn in both hemispheres, in an area within the tropics and the autumn pole between 90 - 100 km (Figure A5). Comparison of FM19 $H_2O$ retrievals shows instead a relative difference of 0% between 40 - 45 km and of +5% between 50 - 60 km. The observed v2.1 FM19 negative bias below 60 km is therefore partly reduced. Then the difference goes down to -15% at 80 km, back up to -5% between 85 - 90 km and then

down again to -30% at 100 km (Figure 6b). Peaks of -40% are observed during northern spring and summer at high latitudes around 100 km altitude (Figure A6). Note that each relative difference value is referred to the mean between the v3.0 and v2.1 concentrations. Temperature profile comparisons, and the zonal means showing differences for the various latitude and seasons, are described below and shown in the Appendix. The retrieved temperature for FM13 is generally lower for v3.0 compared to v2.1, with an absolute difference oscillating between -2.5 and -5 K in the 40 - 90 km altitude range (Figure A7a). FM19

v3.0 temperature is instead generally higher than measured in v2.1, with the absolute difference being equal to +7 K at 40 km and then oscillating between +2.5 and +5 K in the 45 - 90 km altitude range (Figure A7b). For both frequency modes, the relative differences are very low. These differences show that the data set has been improved, since the previous version was affected by a high bias in the case of FM13 and by a generally low bias in the case of FM19, as described in Section 2.2. These improvements will be evaluated further in the next section, where the v3.0 data sets will be compared to other instruments.





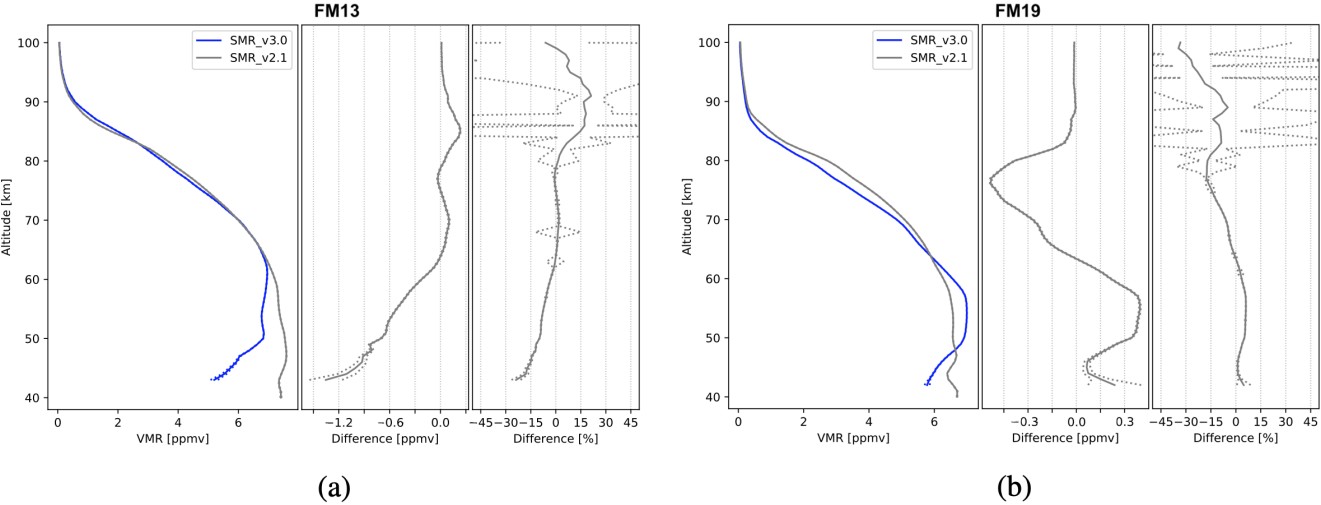

**Figure 6.** Comparison of SMR v3.0 with respect to v2.1 $H_2O$ concentrations, from FM13 (a) and FM19 (b). The data plotted are global averages over the whole time period between February 2001 and April 2019. Each subfigure consists of three panels. Left panels: volume mixing ratios, expressed in ppmv. Center panels: absolute differences, expressed in ppmv. Right panels: relative differences, expressed in percentage. The differences shown are calculated as medians of the single profiles differences. The dashed lines represent the standard deviation of the median which, in some cases, is smaller than the thickness of the profile line, causing the dashed line not to be distinguishable.

## 4 Comparison with other instruments

To evaluate the quality of the new FM13 and FM19 data, in this section we compare the SMR v3.0 $H_2O$ retrievals from these FMs with coincident measurements from other limb-sounding satellite-borne instruments, that is MIPAS, ACE-FTS and MLS. Moreover, we also look at how temperature compares to these instruments.

5   Regarding $H_2O$, measurements are considered coincident if they occur within a maximum temporal separation of 9 hours and maximum spatial separation of 800 km, while for temperature the criteria are 4 hours and 1000 km. The different criteria depend on the temporal variability of the products and on the amount of data available from the comparison instruments. While ACE-FTS and MLS profiles have similar vertical resolutions to SMR, MIPAS profiles are characterised by more coarse resolutions. Therefore, for comparison with MIPAS, SMR profiles have been smoothed with a Gaussian filter characterised

10  by a Full Width at Half Maximum (FWHM) equal to the MIPAS–SMR difference in vertical resolution. Then it is possible to carry out comparisons just by interpolating the profiles over a common 40 - 100 km altitude grid with a 1 km resolution. Denoting with $i$ a couple of coincident measurements, the absolute and relative difference between these profiles are defined, respectively, as:

$$\delta_{abs,i}(z) = x_{SMR} - x_{comp} \tag{8}$$





$$\delta_{rel,i}(z) = \frac{x_{SMR} - x_{comp}}{(x_{SMR} + x_{comp})/2}. \tag{9}$$

where $x_{SMR}$ and $x_{comp}$ are the retrieved $H_2O$ mixing ratios or temperature at altitude $z$ for the coincidence $i$, from SMR and the instrument considered for the comparison, respectively. Measurements done by satellite instruments are in general affected by large uncertainties so, when comparing them, their relative difference is referred to the mean of the two, not to prefer any instrument as a reference (Randall et al., 2003). $\Delta(z)$ being the median of the $N(z)$ differences (absolute or relative) measured at altitude $z$, the dispersion of the measurements is represented by the standard deviation of the median:

$$SEM(z) = \frac{1}{\sqrt{N(z)}} \sqrt{\frac{1}{N(z)-1} \sum_{i=1}^{N(z)} (\delta_i(z) - \Delta(z))^2}. \tag{10}$$

The median is used, instead of the mean, to minimize the impact of outliers data. Below we present the results of the comparisons in form of profiles averaged over the totality of the coincidences, regardless of time or location. Both $H_2O$ and temperature profiles are discussed below, the latter being shown in the appendix. For the sake of clarity, no monthly or seasonal average profiles are shown, but seasonal zonal means of $H_2O$ VMR relative differences and temperature absolute differences are also included in the appendix and discussed.

## 4.1 MIPAS

The Michelson Interferometer for Passive Atmospheric Sounding (MIPAS) on board Envisat performed mid-infrared limb sounding of the atmosphere from June 2002 until April 2012, when contact with the satellite was lost. Envisat was on a 98.55° inclination and 22:00 hrs ascending node sun-syncronous orbit at 800 km altitude. The retrieval products used for comparison in this study are obtained with the retrieval processor developed at the Karlsruhe Institute of Meteorology and Climate Research (IMK) and the Instituto de Astrofisica de Andalucia (IAA) (von Clarmann et al., 2009). The characteristics of the MIPAS V5 datasets being used are summarized in Table 2. The forward model used for Middle Atmosphere (MA) and Upper Atmosphere (UA) modes includes non-LTE effects (Funke et al., 2001). Quality filtering of the data, as indicated in Kiefer and Lossow (2017), has been performed. In March 2004, MIPAS underwent a malfunctioning and was made operative again in January 2005. During the first period, the instrument was being used in full spectral resolution (FR mission) while, in the second period, it was made operative again with a reduced spectral resolution (OR mission) (Oelhaf, 2008).

### 4.1.1 Nominal mode

Both Nominal Mode datasets altogether (from FR and OR periods) are here considered for comparison with SMR. Differences between the two reference data sets (FR and OR) are so small in most regions that they do not spoil the comparison, and only between 45-50 km they reach 10% (not shown). Therefore the OR and FR data are considered as one data set and comparisons are not presented separately for each of them. In particular, for comparisons with SMR FM13, no variations between FR and OR are to be reported since only a small quantity of SMR FM13 measurements have been performed during the period of





**Table 2.** Characteristics of the MIPAS $H_2O$ and temperature datasets used for comparison. Vertical resolutions refer to the observations in the altitude range 40 - 100 km considered in this study.

| Observation Mode | Altitude Range | Product | Vertical Resolution | Spectral Resolution Mode | Time Period | Version |
|---|---|---|---|---|---|---|
| Nominal (NOM) | 10 - 70 km | $H_2O$ | 5 - 15 km | Full Resolution (FR) | July 2002 → March 2004 | V5H_H2O_20 |
| | | T | / | | | / |
| | | $H_2O$ | 5 - 16 km | Optimized Resolution (OR) | January 2005 → April 2012 | V5R_H2O_220 |
| | | T | / | | | / |
| Middle Atmosphere (MA) | 20 - 100 km | $H_2O$ | 4 - 10 km | | | V5R_H2O_522 |
| | | T | 3 - 9 km | | | V5R_T_521 |
| Upper Atmosphere (UA) | 42 - 150 km | $H_2O$ | 3 - 10 km | | | V5R_H2O_622 |
| | | T | 3 - 9 km | | | V5R_T_621 |

MIPAS FR mission (see Figure 3). SMR $H_2O$ average profiles for FM13 and FM19, averaged over all the coincidences found, show different agreement with MIPAS. FM19 relative difference (see Figure 7:b) is very low between 40 and 60 km altitude staying close to 0%. Then the difference increases to reach +10% at 65 km and decreases back to 0% at 70 km. Looking at latitudes and seasons specifically (Figure A11), peaks of +20% are observed around 65 km at high latitudes during local
summer and northern spring; while relative differences of -30% are observed at 70 km during local winter at mid latitudes. FM13 instead (Figure 7:a) presents a major negative difference of -40% at 40 km, decreasing to -10% at 45 km. Between 45 and 70 km, the difference value increase and reaches +20%. The highest positive differences, of +40% are registered between 60 and 70 km during local summer in both hemispheres and during northern spring (Figure A10) at high latitudes. Lossow et al. (2019), from a thorough comparison of satellite datasets, state that at around 65 km MIPAS Nominal $H_2O$ concentration
reaches a negative bias of up to -25%. This might explain the SMR–MIPAS relative differences which we observe around that altitude for both FMs.

### 4.1.2  Middle atmosphere mode

$H_2O$ profile comparison between MIPAS OR Middle Atmosphere mode and SMR FM19 presents a relative difference of -15% at 40 km which increases with altitude to reach 0% at 55 km, and stays roughly constant until 65 km (Figure 8:b).
It then increases to +10% - i.e. 0.4 ppmv in absolute difference - at 70 km and decreases up to -80% (out of shown scale; corresponding to -0.4 ppmv) at 90 km; finally, it increases again to reach -5% at 100 km. For FM13 instead (Figure 8:a), the relative difference has a value of -20% at 40 km and increases to 0% at 60 km. It increases up to around 20% (corresponding to 1 ppmv absolute difference) at 70 km and then it decreases to -100% (- 0.3 ppmv) around 90 km and goes up again to around -10% at 100 km. The differences observed for both FMs between 40 and 60 km are consistent with the MIPAS MA high bias
reported by Lossow et al. (2019) at those altitudes. For both FMs, peaks of - 150% are observed for all seasons between 90 and 100 km at low latitudes. At the same altitudes, smaller differences during polar winter are observed. This is probably explained


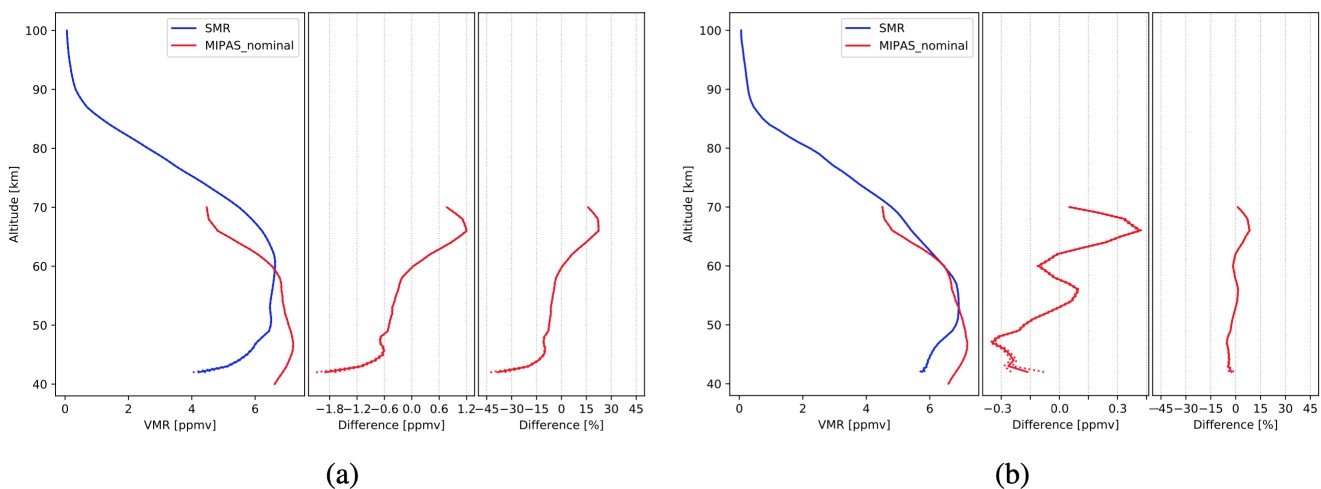

**Figure 7.** Comparison of SMR $H_2O$ concentrations, from FM13 (a) and FM19 (b), with the ones from MIPAS Nominal Mode retrievals. The data plotted are global averages over the whole time between periods indicated in Table 2. Figures characteristics are the same as in Figure 6.

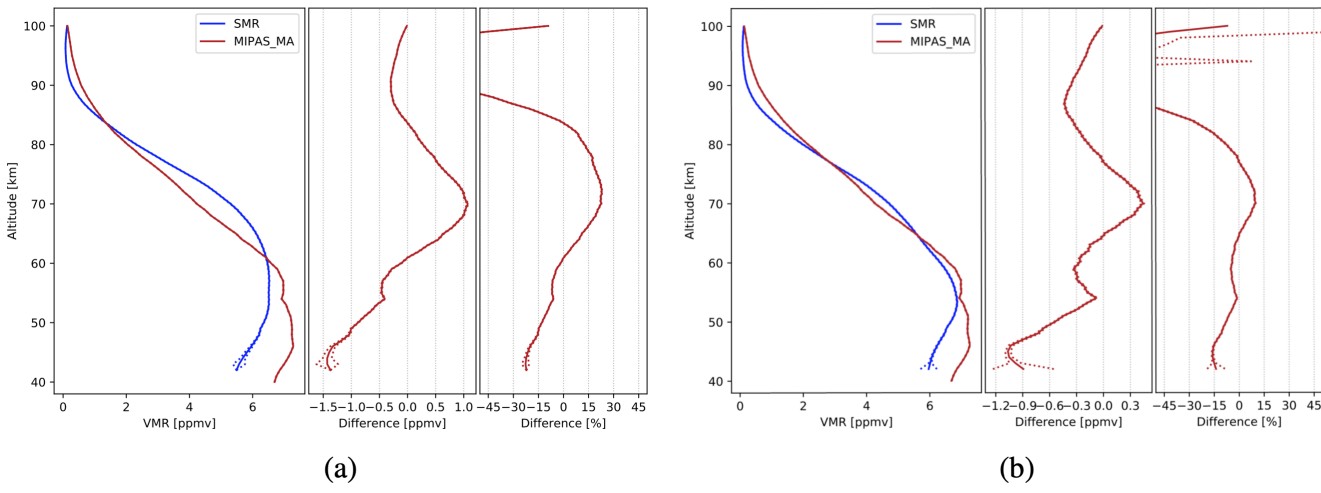

**Figure 8.** Comparison of SMR $H_2O$ concentrations, from FM13 (a) and FM19 (b), with the ones from MIPAS Middle Atmosphere Mode retrievals. The data plotted are global averages over the whole time periods indicated in Table 2. Figures characteristics are the same as in Figure 6.





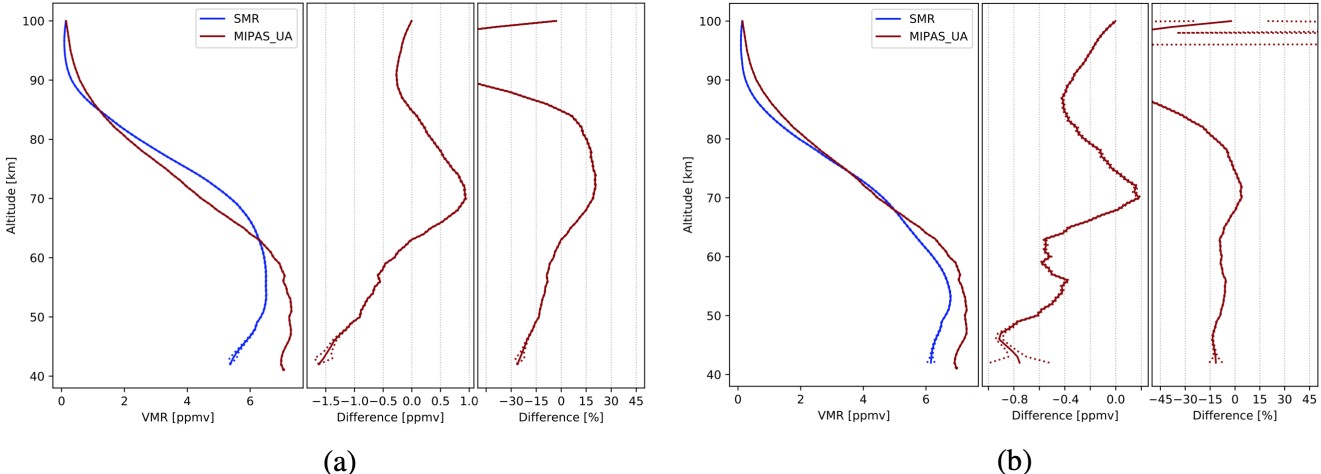

**Figure 9.** Comparison of SMR $H_2O$ concentrations, from FM13 (a) and FM19 (b), with the ones from MIPAS Upper Atmosphere Mode retrievals. The data plotted are global averages over the whole time periods indicated in Table 2. Figures characteristics are the same as in Figure 6.

by non-LTE effects being less important there (Figures A12 and A13). Note that very high values in $H_2O$ relative difference are to be expected at high altitudes due to the extremely low concentrations in that region. Temperature absolute differences are close to 0 K between 40 and 80 km and decrease down to -45 K at higher altitudes, for both FMs (Figure A20).

### 4.1.3 Upper atmosphere mode

$H_2O$ average relative differences between SMR and MIPAS OR Upper Atmosphere profiles present a value of -30% at 40 km for FM13 (Figure 9:a) which gets smaller with altitude to reach 0% at 60 km. The value keeps increasing with altitude until 70 km where it reaches +20% (1 ppmv), it stays roughly constant until 80 km, decreases to about -100% (-0.3 ppmv) at 90 km, and finally increases back to 0% at 100 km. Relative difference regarding FM19 (Figure 9:b) is equal to -15% at 40 km; it oscillates between -10% and -5% until 65 km and reaches 0% at 70 km; it then decreases to reach -100% (-0.4 ppmv) at 90 km
and increases back to 0% at 100 km. Peaks of -150% are registered between 90 and 100 km at low latitudes during all seasons for both FMs (Figures A14 and A15), as well as a peak of +90% during southern winter at high latitudes. Temperature absolute difference is similar to what was observed in the comparison with the Middle Atmosphere mode (Figure A23).

### 4.2 ACE-FTS

The Fourier Transform Spectrometer (FTS), an instrument which is part of the Canadian-led Atmospheric Chemistry Experi-
ment (ACE), was launched on board Scisat-1 on 12 August 2003 and it is still operating today. The satellite travels at 650 km altitude, with an orbit characterised by a 74° inclination. The instrument measures $H_2O$ concentration between 5 and 100 km, and temperature up till 125 km, with a 3 - 4 km vertical resolution. For comparison, we use here the ACE-FTS v3.6 dataset





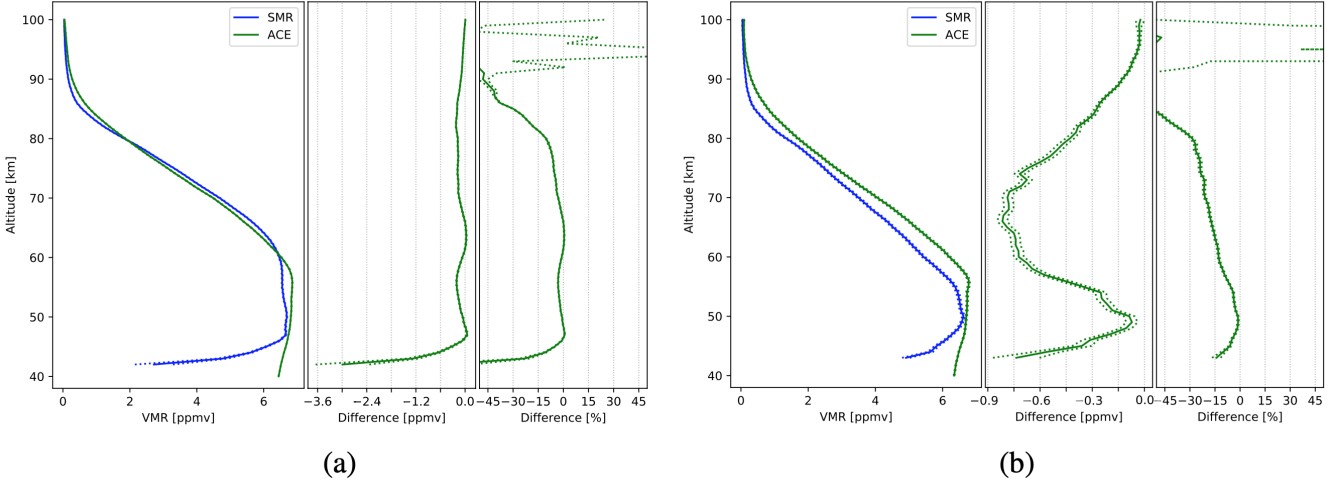

**Figure 10.** Comparison of SMR $H_2O$ concentrations, from FM13 (a) and FM19 (b), with the ones from ACE-FTS retrievals. The data plotted are global averages over the whole time between February 2004 and April 2019. Figures characteristics are the same as in Figure 6.

(Sheese et al., 2017), quality filtered as indicated by the instrument team (Sheese et al., 2015).

SMR–ACE $H_2O$ profile comparison, with regards to FM13 (Figure 10:a), shows a -70% relative difference at 40 km, then the value goes steeply to 0% and stays almost constant between 45 and 80 km altitude. Between 80 and 100 km the relative
difference value goes down and reaches values of -140% (corresponding to absolute differences in the order of -0.01 ppmv). For FM19 (Figure 10:b), the measured relative difference is -15% below 45 km and reaches 0% at 50 km; it then decreases slowly with altitude until 80 km where it is equal to -30% (-0.40 ppmv absolute difference). Between 80 and 100 km it decreases more rapidly to about -60% (corresponding to an absolute difference in the order of -0.01 ppmv). Regarding temperature (Figure A26), FM13 absolute difference stays between 0 and 4 K until 80 km altitude, and at higher altitudes it oscillates between lower
values within 0 and - 16 K. For FM19 instead, it assumes values between 0 and 7 K until 50 km, then it slowly decreases up to -15 K at 90 km, and between 90 and 100 km the difference is characterised by considerably lower values with a minimum of -60 K.

### 4.3 MLS

The Microwave Limb Sounder (MLS) operates on board the Aura satellite since 15 July 2004, on a 705 km sun-synchronous
orbit characterized by a 98°inclination and a 13:45 hrs ascending node. Aura/MLS observes between 118 GHz and 2.5 THz with a 1.5 - 3 km vertical resolution (Schoeberl et al., 2006). We use MLS measurements of temperature, as well as $H_2O$ concentration, from the v5 dataset to which the recommended quality filtering has been applied (Livesey et al., 2020).

Comparing SMR $H_2O$ profiles from FM13 with MLS (Figure11:a), we observe a relative difference of -30% at 40 km which rapidly goes up to 0% at 45 km and then remains constant until 65 km altitude. The value then decreases to reach -30% (-0.6





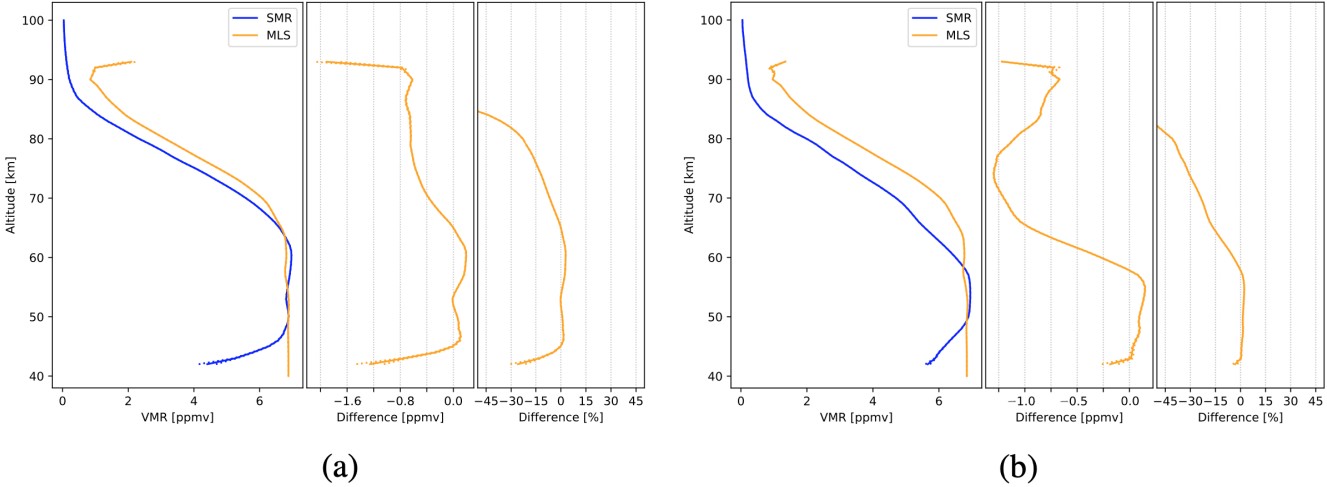

(a)                (b)

**Figure 11.** Comparison of SMR $H_2O$ concentrations, from FM13 (a) and FM19 (b), with the ones from MLS retrievals. The data plotted are global averages over the whole time between July 2004 and April 2019. Figures characteristics are the same as in Figure 6.

ppmv absolute difference) at 80 km. Above 80 km, it decreases rapidly with altitude and reaches -140% (-2 ppmv) at 95 km. Regarding comparison with FM19 profiles, the difference is equal to 0% between 40 and 60 km. Between 60 and 80 km it goes down to -30% (-1 ppmv) and then quickly decreases with altitude down to - 165% (-1.25 ppmv) at 95 km. Peaks of -150% are observed at 90 km during local winter and autumn in both hemispheres for both FMs (Figures A18 and A19). Temperature

absolute difference for FM13 (Figure A29a) is equal to 8 K at 40 km, goes to 0 K at 45 km, increases back up to 8 K at 60 km. The value is constant with altitude between 60 - 70 km. The difference then decreases down to -2 K around 90 km, goes back to 8 K at 95 km and decreases to -3 K at 100 km. FM19 shows a decrease in temperature from 10 K at 40 km to 5 K at 45 km (Figure A29b). The values stays constant between 45 and 55 km and then decrease with altitude to reach -5 K around 90 km. Finally, an absolute difference of 0 K is reached around 100 km.

## 5 Summary and conclusions

The previous version (v2.1) of SMR FM13 and FM19 $H_2O$ and temperature products presented large biases compared to other instruments. In particular, FM19 $H_2O$ presented a bias between ±20% between 40 and 80 km, while FM13 concentrations were around 10% higher than ACE-FTS and MLS between 40 and 60 km, and around 20% higher than MIPAS in the same altitude range. Above 80 km, both FMs presented differences greater than -50%. FM 19 temperature had a bias of around -5

– -10 K between 60 and 80 km; and FM 13 temperature bias was equal to around +10 K between 40 and 80 km. Both FMs were characterised by very high negative biases at high altitudes. After investigating different possible causes, we identified the origin of these biases in the mistaking of spurious signal from sideband leakage for actual signal of interest. This was caused by having overestimated the capability of image band suppression performed by the frontends employed for measurements with

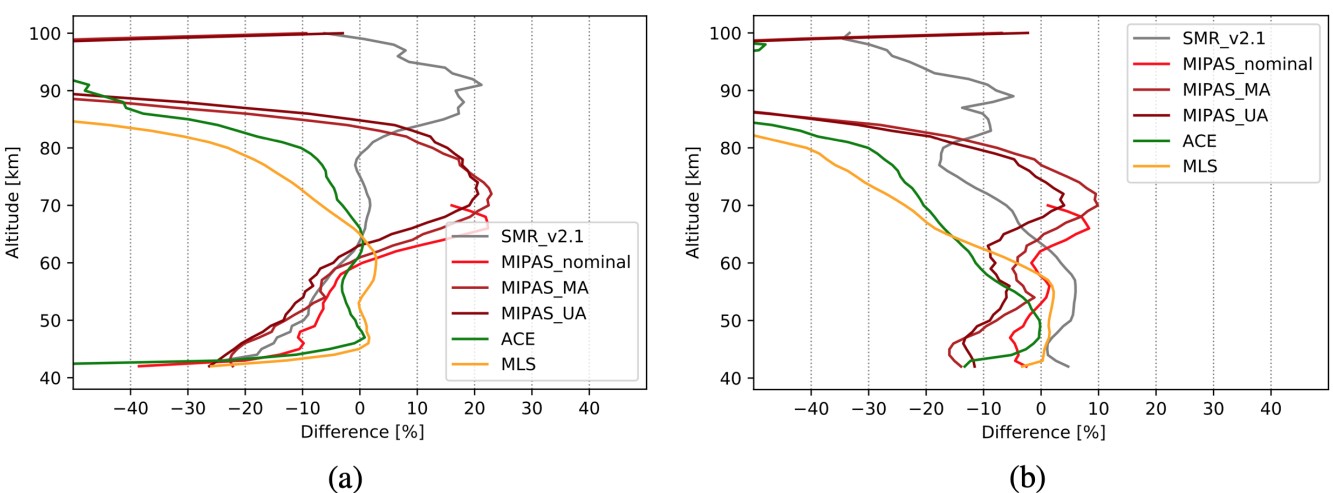

**Figure 12.** Summary of relative differences of SMR v3.0 $H_2O$ concentrations with respect to SMR v2.1 ones as well as those retrieved from measurements by all other instruments considered in this study. For the sake of clarity, errors are not shown. Panel a: FM13 comparison. Panel b: FM19 comparison.

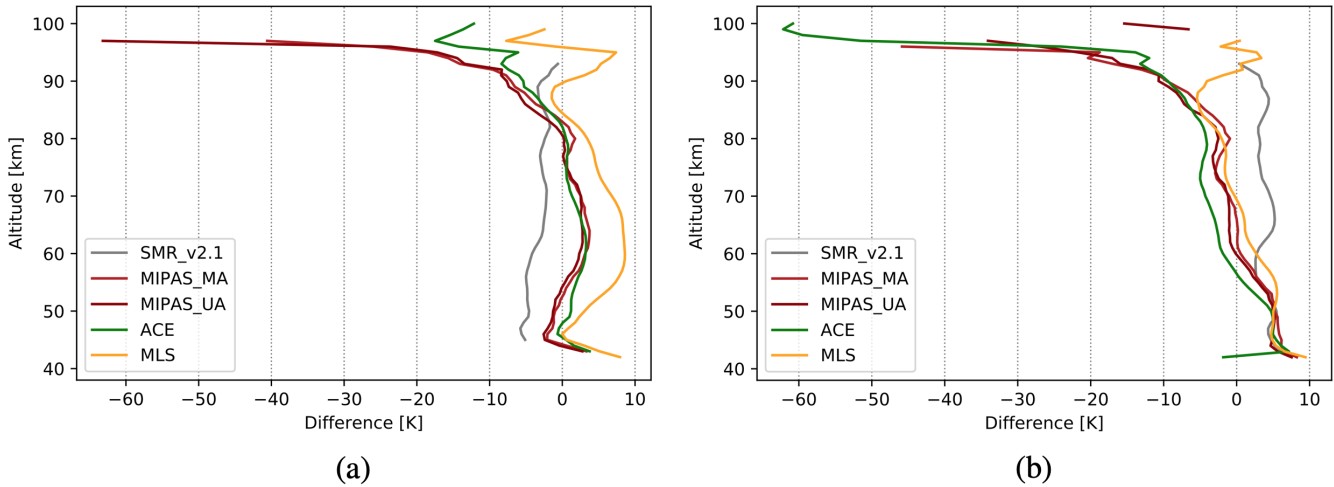

**Figure 13.** Summary of absolute differences of SMR v3.0 temperatures with respect to SMR v2.1 ones as well as those retrieved from measurements by all other instruments considered in this study. For the sake of clarity, errors are not shown. Panel a: FM13 comparison. Panel b: FM19 comparison.

these FMs. A lower suppression has therefore been assumed and retrievals with the new settings have been performed. This resulted in a new dataset (v3.0) covering 18 years of observations from 40 to 100 km altitude, across all latitudes. Time series of $H_2O$ concentration and temperature show temporal variation patterns which are consistent with the current knowledge on mesospheric water vapour and temperature, like the signature of semi-annual oscillation and annual cycle for example. The





validation study, performed by comparing SMR observations with independent satellite measurements from MIPAS, ACE-FTS and MLS, shows that globally averaged SMR v3.0 FM13 $H_2O$ concentrations (Figure 12a) present relative differences within $\pm20\%$ between 45 and 80 km altitude. In particular SMR is in very good agreement with ACE-FTS and MLS in this altitude range, with relative differences within 0% and -5%. Relative differences between v3.0 FM19 and all instruments are within

$\pm20\%$ between 40 and 70 km. In particular, differences with regards to the different MIPAS observation modes are within $\pm10\%$ up to 80 km (Figure 12b). For both FMs, outside the above mentioned altitude ranges, relative differences reach highly negative values, to a minimum of -140%. This is to be expected due to the fact that $H_2O$ concentration values are very low at higher altitudes. It can also be seen that for SMR v3.0 FM19, between 40 and 60 km, there is a general reduction of the relative difference with respect to all other instruments, compared to v2.1. This consists of a few percent with respect to MLS

and reaches 10-15% with respect to MIPAS MA and UA modes. Moreover, temperature shows an improvement of about 5 K in absolute difference at all observed altitudes with respect to the previous version, for both FMs (Figure 13). Only FM19 in the 40 - 60 km altitude range is an exception, where v2.1 agreed better with the other instruments. Temperature from v3.0 FM13 agrees very well with all MIPAS modes and with ACE-FTS between 40 and 85 km, presenting absolute differences within $\pm3$ K. In the same altitude range, SMR-MLS difference however oscillates between +8 K and 0 K. SMR v3.0 FM19

temperature absolute difference from all other instruments is equal to +8 K at 40 km and gradually decreases to reach - 8 K at 85 km. For both FMs, altitudes above 85 km are characterised by lower absolute differences with respect to almost all instruments, reaching - 60 K at 100 km. SMR–MLS difference is an exception, with a value between $\pm8$ K at these altitudes.

The global mesospheric water vapour and temperature data from Odin/SMR have been reprocessed, leading to a significant improvement of the L2 products. The data sets are available to the scientific community at https://odin.rss.chalmers.se/dataaccess.

They represent valuable tools for the study of middle atmospheric chemistry and dynamics, as well as for trend studies, given their important time coverage (more than 18 years).

**Appendix A**







**Figure A1.** Time series of FM13 $H_2O$ volume mixing ratios measured by SMR for different latitude bands. The white areas indicate periods and altitudes at which the number of measurements in the given latitude band is lower than 10. The ticks on the x-axis correspond to the beginning of each year.





**Figure A2.** Time series of FM13 temperature measured by SMR for different latitude bands. The white areas indicate periods and altitudes at which the number of measurements in the given latitude band is lower than 10. The ticks on the x-axis correspond to the beginning of each year.





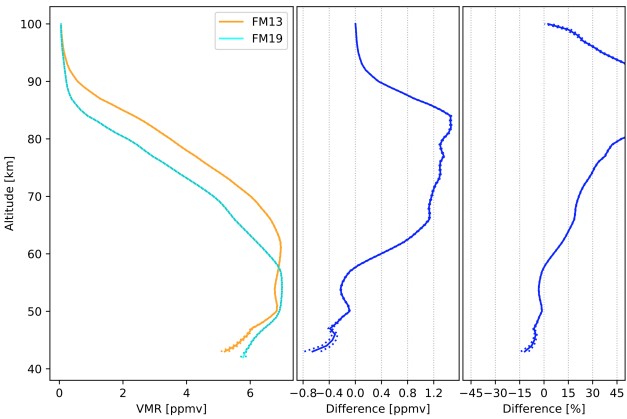

**Figure A3.** Absolute differences (left) and relative differences (right) between SMR v3.0 FM13 and FM19 $H_2O$ concentrations. The data plotted are global averages over the whole time period between February 2001 and April 2019.

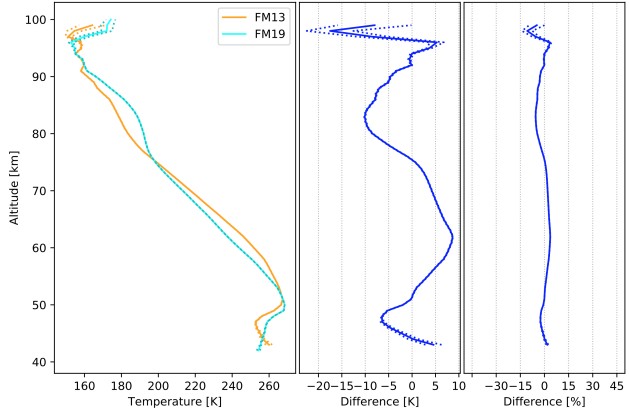

**Figure A4.** Absolute differences (left) and relative differences (right) between SMR v3.0 FM13 and FM19 temperatures. The data plotted are global averages over the whole time period between February 2001 and April 2019.



**Figure A5.** Seasonal zonal means of $H_2O$ FM13 SMR v3.0–v2.1 relative differences averaged over the whole time period between February 2001 and April 2019. The seasons are intended as astronomical seasons, i.e. starting each at the respective solstice or equinox.



**Figure A6.** Seasonal zonal means of $H_2O$ FM19 SMR v3.0–v2.1 relative differences averaged over the whole time period between February 2001 and April 2019. The seasons are intended as astronomical seasons, i.e. starting each at the respective solstice or equinox.



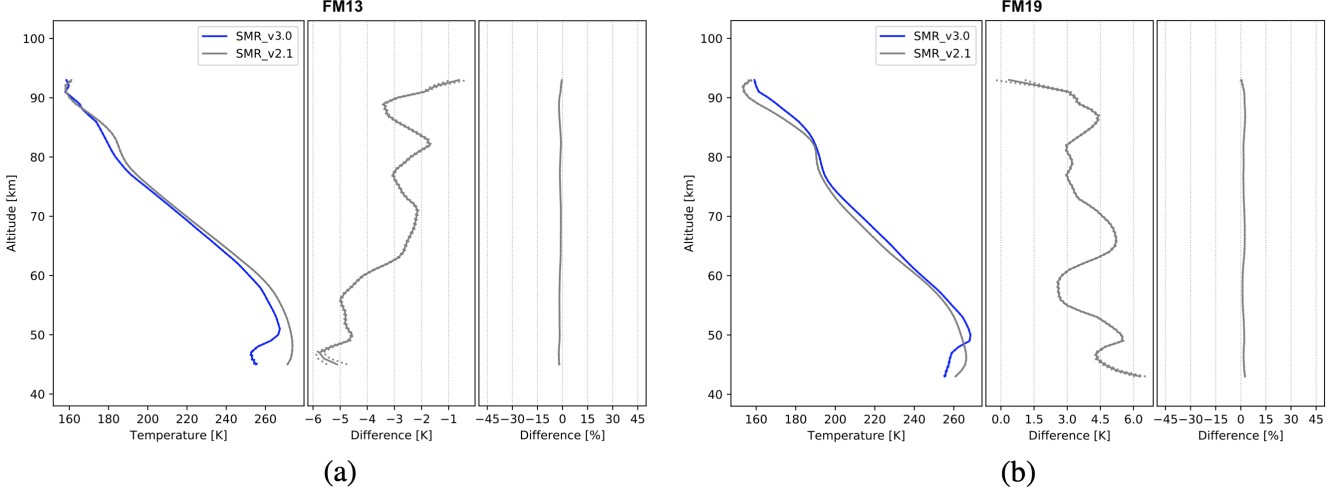

**Figure A7.** Comparison of SMR v3.0 and v2.1 temperatures from FM13 (a) and FM19 (b). The data plotted are global averages over the whole time period between February 2001 and April 2019. Figures characteristics are the same as in Figure 6.



**Figure A8.** Seasonal zonal means of temperature FM13 SMR v3.0–v2.1 relative differences averaged over the whole time period between February 2001 and April 2019. The seasons are intended as astronomical seasons, i.e. starting each at the respective solstice or equinox.

**Figure A9.** Seasonal zonal means of temperature FM19 SMR v3.0–v2.1 relative differences averaged over the whole time period between February 2001 and April 2019. The seasons are intended as astronomical seasons, i.e. starting each at the respective solstice or equinox.

**Figure A10.** Seasonal zonal means of $H_2O$ FM13 SMR–MIPAS Nominal relative differences averaged over the time period indicated in Table 2. The seasons are intended as astronomical seasons, i.e. starting each at the respective solstice or equinox.

**Figure A11.** Seasonal zonal means of $H_2O$ FM19 SMR–MIPAS Nominal relative differences averaged over the time period indicated in Table 2. The seasons are intended as astronomical seasons, i.e. starting each at the respective solstice or equinox.



**Figure A12.** Seasonal zonal means of $H_2O$ FM13 SMR–MIPAS Middle Atmosphere relative differences averaged over the time period indicated in Table 2. The seasons are intended as astronomical seasons, i.e. starting each at the respective solstice or equinox.

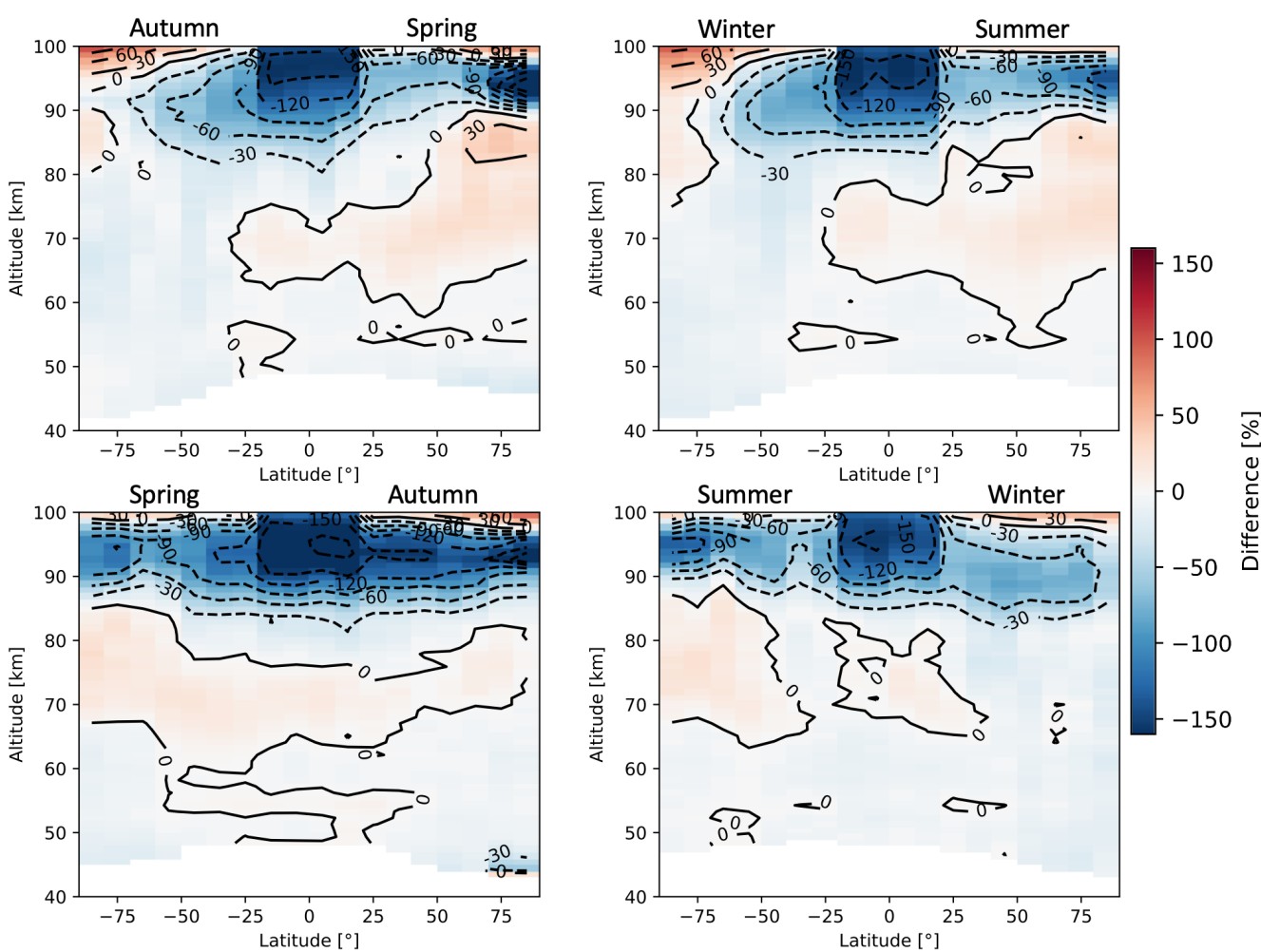

**Figure A13.** Seasonal zonal means of $H_2O$ FM19 SMR–MIPAS Middle Atmosphere relative differences averaged over the time period indicated in Table 2. The seasons are intended as astronomical seasons, i.e. starting each at the respective solstice or equinox.

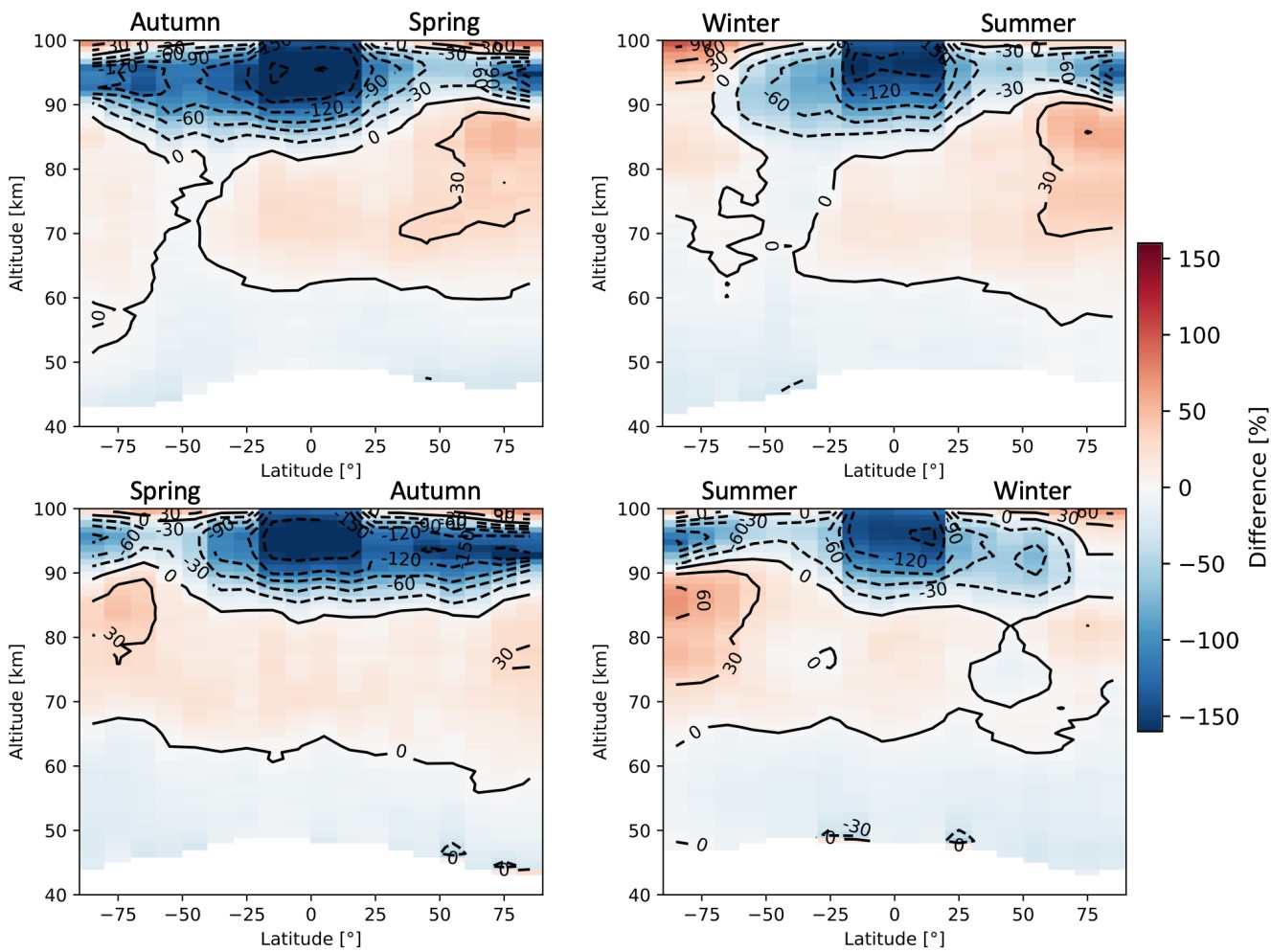

**Figure A14.** Seasonal zonal means of $H_2O$ FM13 SMR–MIPAS Upper Atmosphere relative differences averaged over the time period indicated in Table 2. The seasons are intended as astronomical seasons, i.e. starting each at the respective solstice or equinox.





**Figure A15.** Seasonal zonal means of $H_2O$ FM19 SMR–MIPAS Upper Atmosphere relative differences averaged over the time period indicated in Table 2. The seasons are intended as astronomical seasons, i.e. starting each at the respective solstice or equinox.

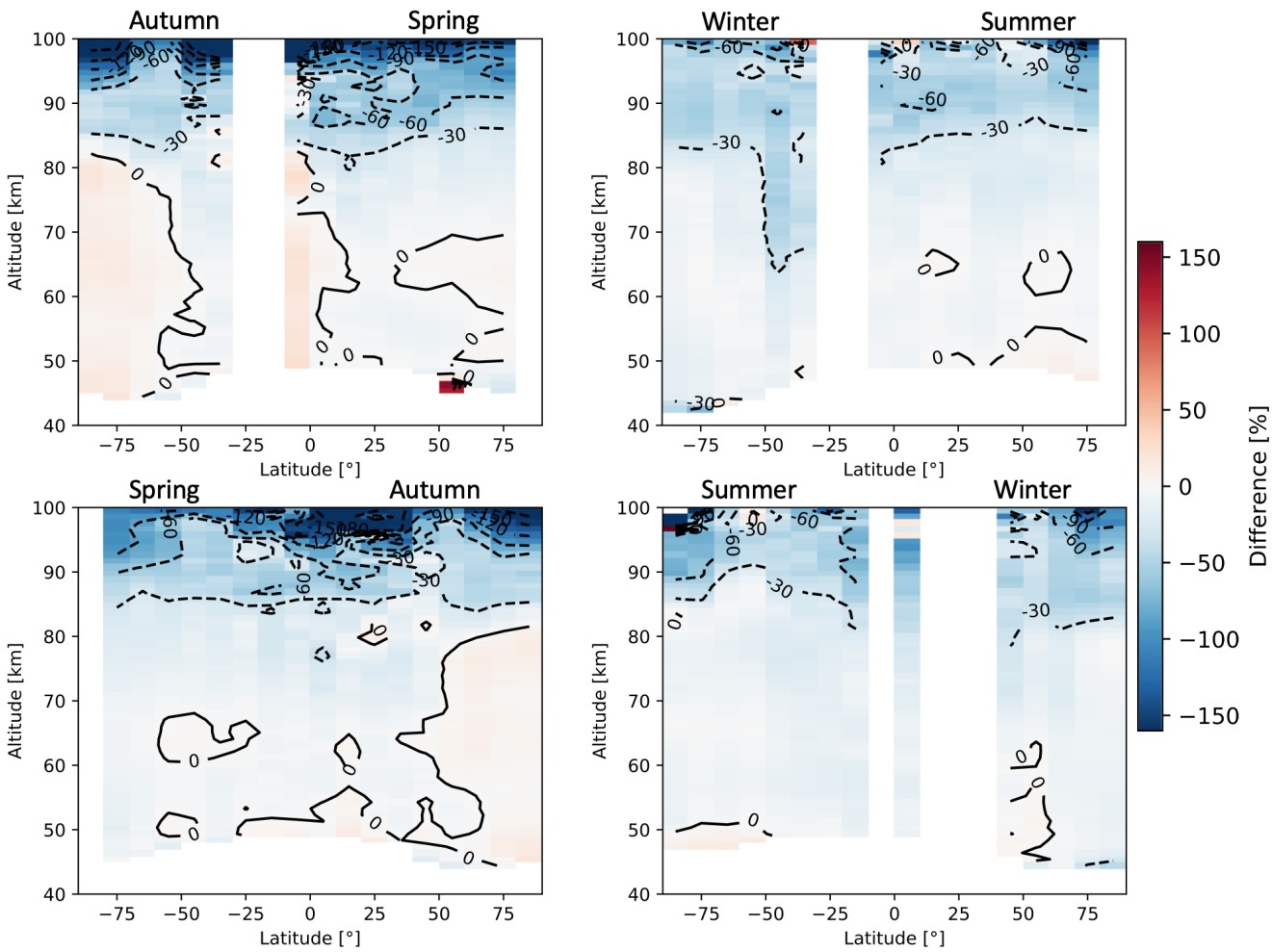

**Figure A16.** Seasonal zonal means of $H_2O$ FM13 SMR–ACE relative differences averaged over the time period between February 2004 and April 2019. The seasons are intended as astronomical seasons, i.e. starting each at the respective solstice or equinox.

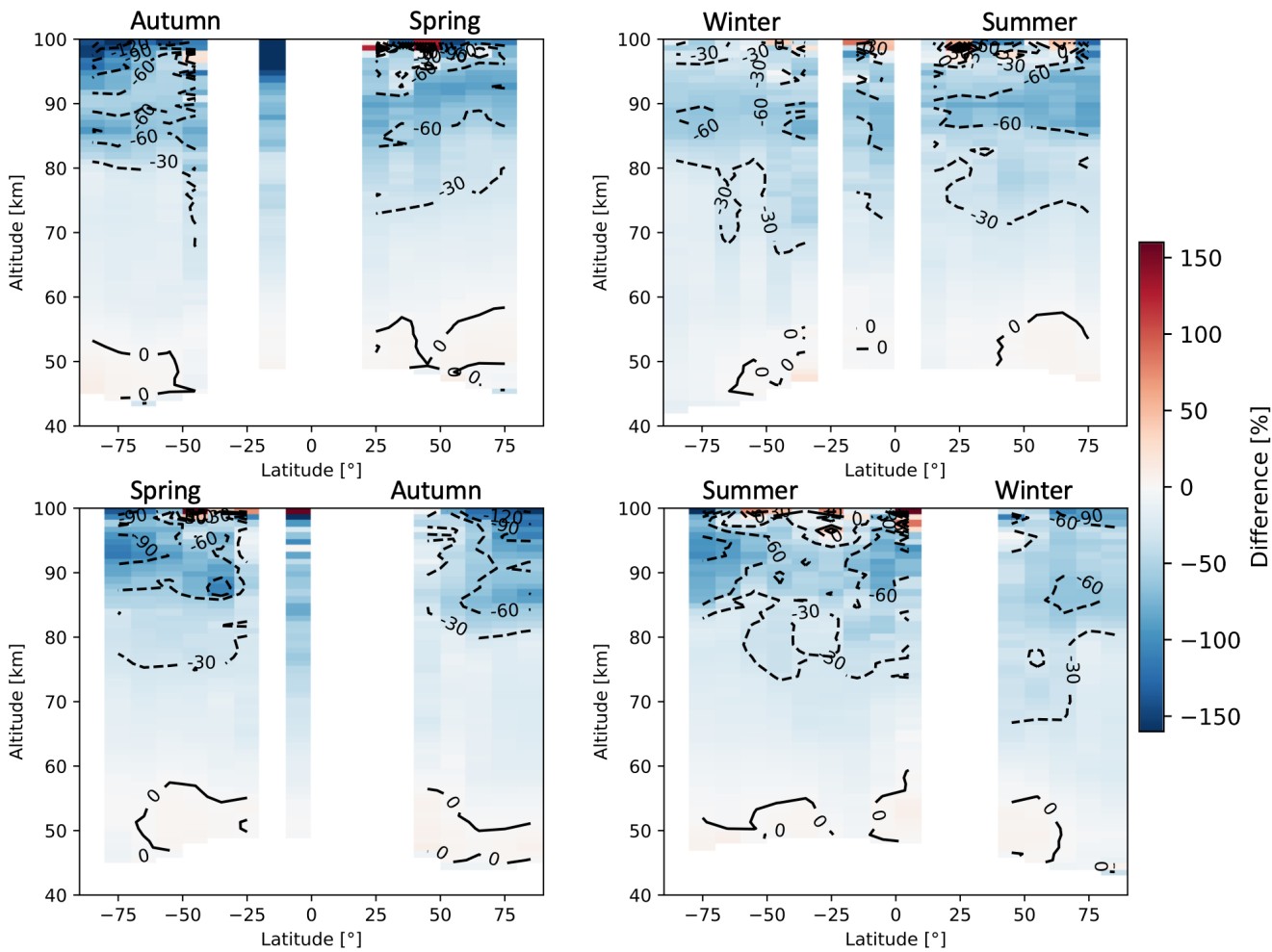

**Figure A17.** Seasonal zonal means of $H_2O$ FM19 SMR–ACE relative differences averaged over the time period between February 2004 and April 2019. The seasons are intended as astronomical seasons, i.e. starting each at the respective solstice or equinox.

**Figure A18.** Seasonal zonal means of $H_2O$ FM13 SMR–MLS relative differences averaged over the time period between July 2004 and April 2019. The seasons are intended as astronomical seasons, i.e. starting each at the respective solstice or equinox.

**Figure A19.** Seasonal zonal means of $H_2O$ FM19 SMR–MLS relative differences averaged over the time period between July 2004 and April 2019. The seasons are intended as astronomical seasons, i.e. starting each at the respective solstice or equinox.



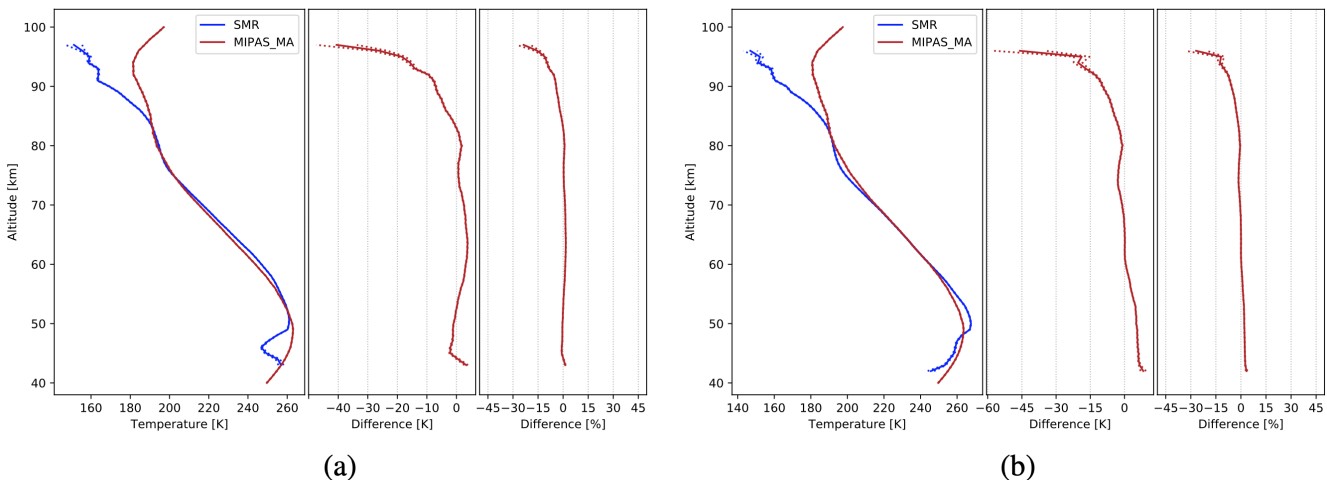

(a)               (b)

**Figure A20.** Comparison of SMR temperatures, from FM13 (a) and FM19 (b), with the ones from MIPAS Middle Atmosphere Mode retrievals. The data plotted are global averages over the whole time periods indicated in Table 2. Figures characteristics are the same as in Figure 6.

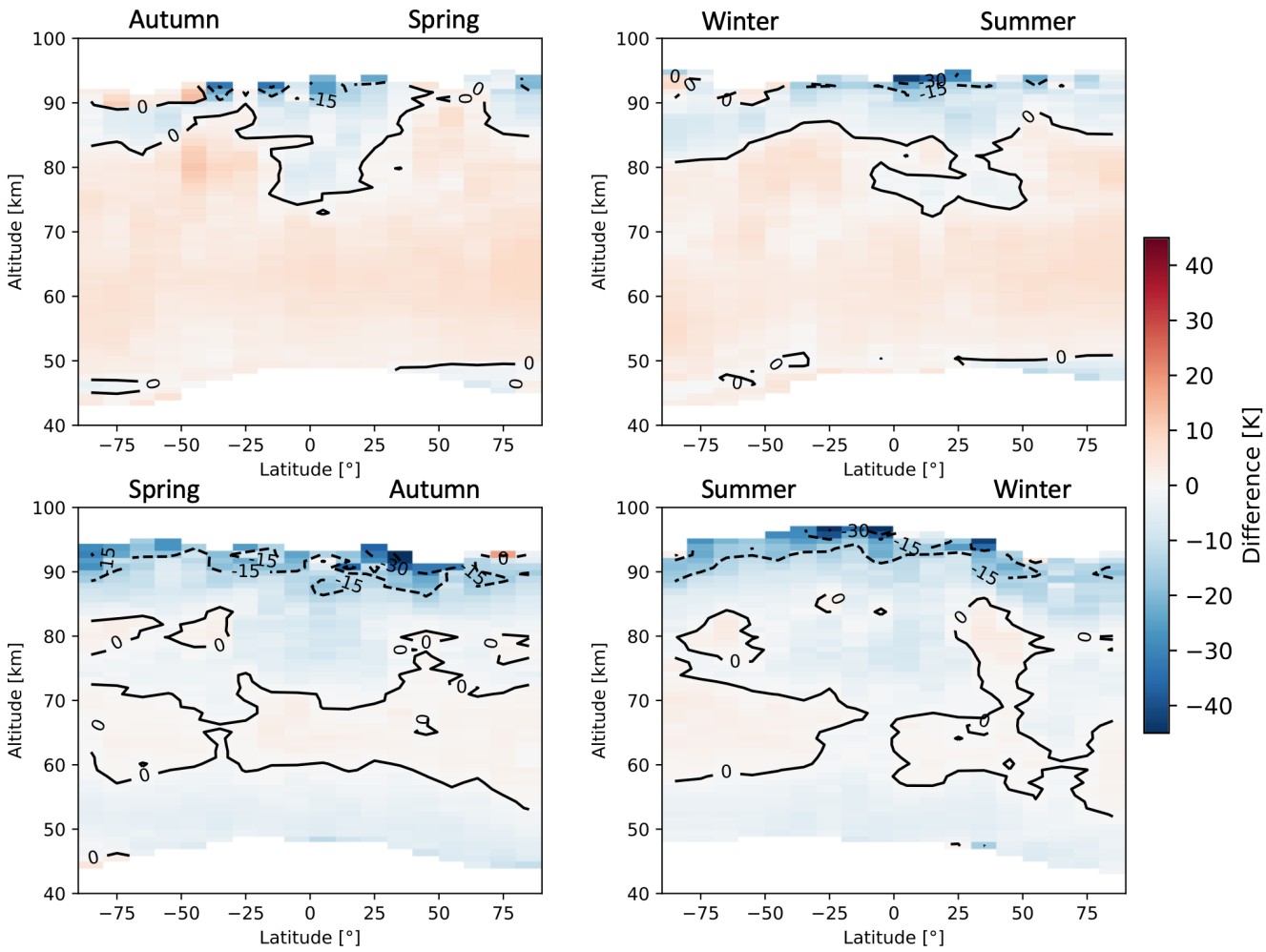

**Figure A21.** Seasonal zonal means of temperature FM13 SMR–MIPAS Middle Atmosphere absolute differences averaged over the time period indicated in Table 2. The seasons are intended as astronomical seasons, i.e. starting each at the respective solstice or equinox.



**Figure A22.** Seasonal zonal means of temperature FM19 SMR–MIPAS Middle Atmosphere absolute differences averaged over the time period indicated in Table 2. The seasons are intended as astronomical seasons, i.e. starting each at the respective solstice or equinox.

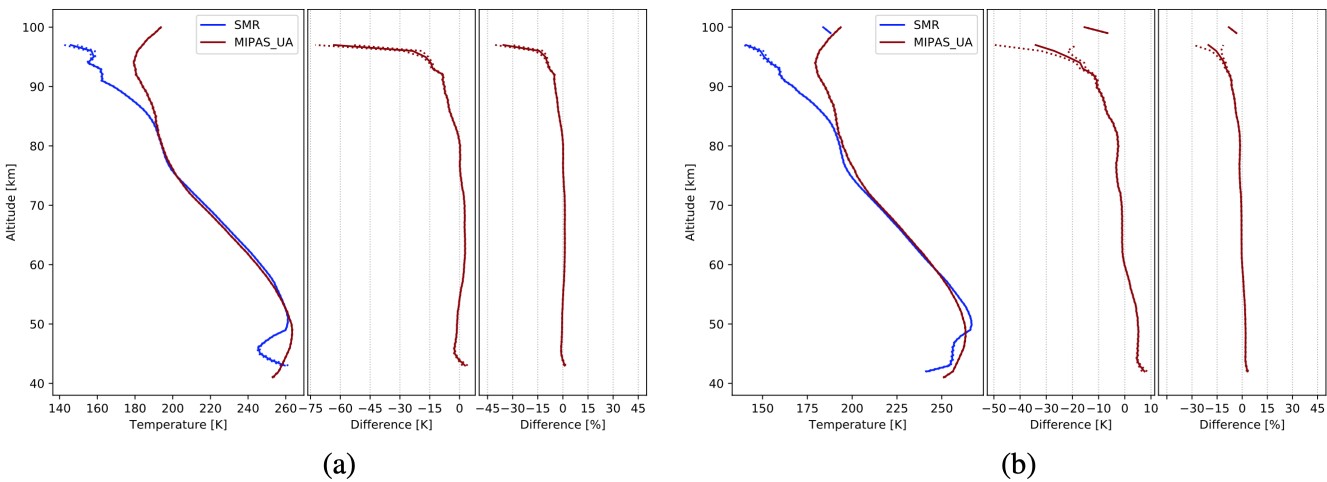

(a)                                     (b)

**Figure A23.** Comparison of SMR temperatures, from FM13 (a) and FM19 (b), with the ones from MIPAS Upper Atmosphere Mode retrievals. The data plotted are global averages over the whole time periods indicated in Table 2. Figures characteristics are the same as in Figure 6.

**Figure A24.** Seasonal zonal means of temperature FM13 SMR–MIPAS Upper Atmosphere absolute differences averaged over the time period indicated in Table 2. The seasons are intended as astronomical seasons, i.e. starting each at the respective solstice or equinox.

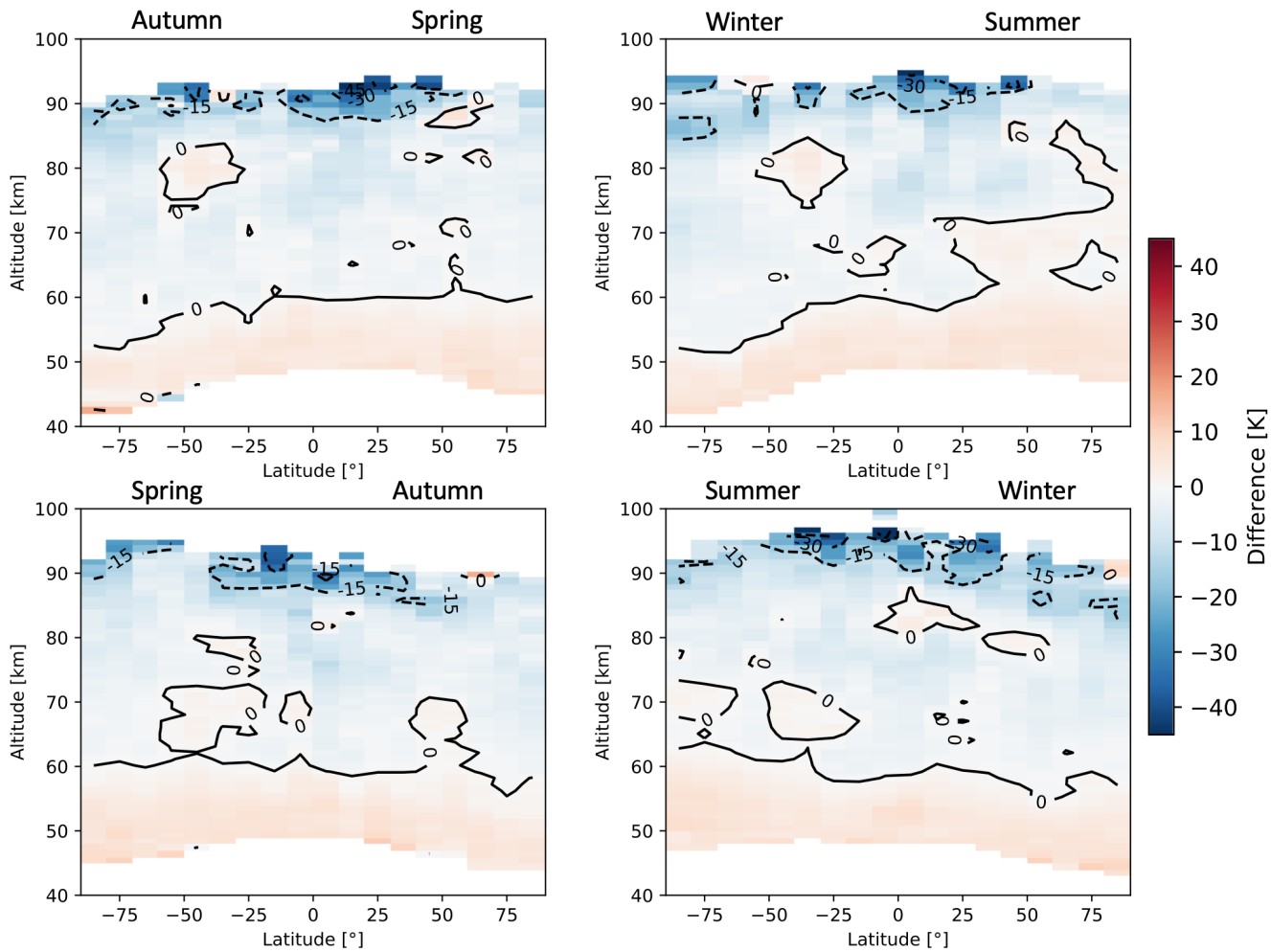

**Figure A25.** Seasonal zonal means of temperature FM19 SMR–MIPAS Upper Atmosphere absolute differences averaged over the time period indicated in Table 2. The seasons are intended as astronomical seasons, i.e. starting each at the respective solstice or equinox.





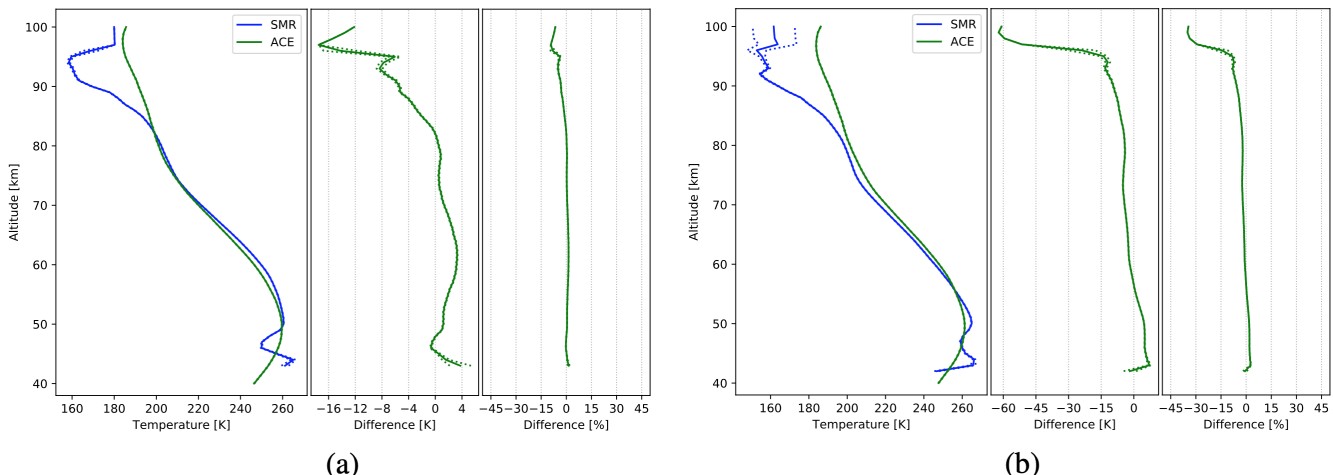

(a)                                                                (b)

**Figure A26.** Comparison of SMR temperatures, from FM13 (a) and FM19 (b), with the ones from ACE-FTS retrievals. The data plotted are global averages over the time period between February 2004 and April 2019. Figures characteristics are the same as in Figure 6.



**Figure A27.** Seasonal zonal means of temperature FM13 SMR–ACE absolute differences averaged over the time period between February 2004 and April 2019. The seasons are intended as astronomical seasons, i.e. starting each at the respective solstice or equinox.



**Figure A28.** Seasonal zonal means of temperature FM19 SMR–ACE absolute differences averaged over the time period between February 2004 and April 2019. The seasons are intended as astronomical seasons, i.e. starting each at the respective solstice or equinox.



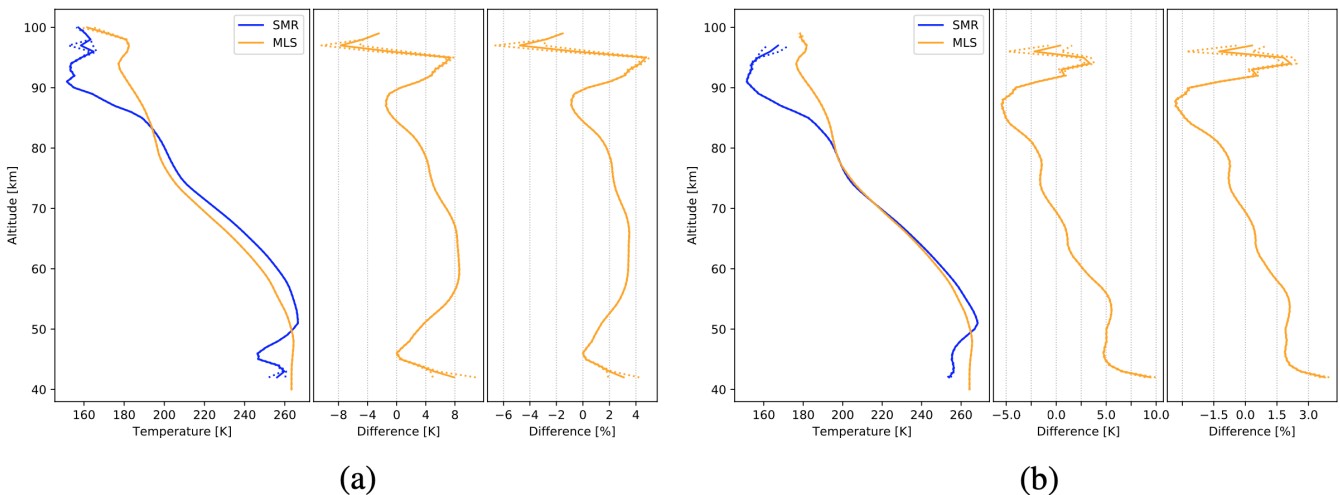

**Figure A29.** Comparison of SMR temperatures, from FM13 (a) and FM19 (b), with the ones from MLS retrievals. The data plotted are global averages over the time period between July 2004 and April 2019. Figures characteristics are the same as in Figure 6.



**Figure A30.** Seasonal zonal means of temperature FM13 SMR–MLS absolute differences averaged over the time period between July 2004 and April 2019. The seasons are intended as astronomical seasons, i.e. starting each at the respective solstice or equinox.



**Figure A31.** Seasonal zonal means of temperature FM19 SMR–MLS absolute differences averaged over the time period between July 2004 and April 2019. The seasons are intended as astronomical seasons, i.e. starting each at the respective solstice or equinox.



*Data availability.* Data availability. Odin/SMR v3.0 L2 data are publicly accessible at http://odin.rss.chalmers.se/level2 (last access: 16 November 2020); MIPAS IMK/IAA L2 data (both NOM and MA/UA) can be downloaded upon registration at http://www.imk-asf.kit.edu/english/308.php (last access: 16 November 2020); ACE-FTS L2 data are available upon request at https://databace.scisat.ca/l2signup.php (last access: 16 November 2020); MLS v5 $H_2O$ L2 data are available at https://doi.org/10.5067/Aura/MLS/DATA2508 (Lambert et al.,
5   2020) and temperature L2 data are available at https://doi.org/10.5067/Aura/MLS/DATA2520 (Schwartz et al., 2020).

*Acknowledgements.* The Chalmers team acknowledges support from the Swedish National Space Agency (Dnr 88/14 and 72/17). Odin is a Swedish-led satellite mission, and is also part of the European Space Agency's (ESA) third party mission programme. The reprocessing of the SMR data was supported by ESA (MesosphEO and Odin/SMR reprocessing projects). The Atmospheric Chemistry Experiment is a Canadian-led mission mainly supported by the Canadian Space Agency. Work at the Jet Propulsion Laboratory, California Institute of
10  Technology, was carried out under a contract with the National Aeronautics and Space Administration.





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
