# Peer review of "Improvement of Odin/SMR water vapour and temperature measurements and validation of the obtained data sets"

_Atmospheric Measurement Techniques, 2021_

## Referee Comment (RC2)

[referee-annotated manuscript omitted]

---

## Author Comment (AC1)

We would like to thank the referees for their constructive comments and suggestions, that helped us improving the quality of our paper. Our detailed replies are included below.

**RC1**

The Grieco manuscript presents the updated retrieval of water vapor and temperature from Odin/SMR by fixing the sideband leakage issue. Validation against other satellite datasets were carried out. Improvement of data in this new version has been achieved.

The paper is well written with significant results. I recommend its pubication after a minor revision.

Major Comments:

1. page 3: line 10-line 14, References for other satellite measurements are not representative. For example, MLS, SABER, SOFIE and ACE-FTS have inappropriate citations. We have changed SABER reference to Dawkins et al. (2018) (https://doi.org/10.1029/2018JD028742) and SOFIE to Stevens et al. (2012) (https://doi.org/10.1029/2012JD017689). We believe MLS and ACE-FTS citations are correct, as they are relative to validation studies.

2. page 5, line 7: "The a priori for water vapour instead...." More informaiton is needed. What kind of measurements at Bordeaux Observatory? This a priori data set was compiled a long time ago, in the beginning of the Odin mission and was not accurately documented. However, the corresponding data is provided for each profile in the v3.0 data files and available at http://odin.rss.chalmers.se/level2.

3. Figures should be descibed in sequence. For example, page 6, line 7 "Figures 12 and 13" are callled too early. We need to call those figures early, because they also show the biases in the SMR v2.1 version (in addition to the relative differences between SMR v3.0 and other instruments) which we talk about in Section 2.2. An alternative to that would be to show very similar plots that only show v2.1 biases already in Section 2.2. We chose not to do this as these plots would be almost a duplicate of the ones in the Conclusions section.

4. Structure of the paper. Many figures are put in the appendix, but are discussed in the main text. Fpr example, page 11, first paragraph. Need to rearrange. We indeed decided to include some of the discussed figures in the appendix instead of keeping all of them in the main paper in order to avoid having too many figures. We had to prioritize some figures over others of secondary importance. We think that such a structure is preferable for the sake of clarity.

5. page 12, line 5 "Regarding H2O, measurements are considered coincident...., while for temperature ......" the separation of time seems too long: 9 hours and 4 hours. Tides will be mixed in the comparison. Performing tests with stricter time coincidence criteria proved not to sensibly change the shape of the median difference profiles, suggesting that tides don't have a significant effect in the presented comparisons. This information has been added to the text.

6. Contours in the appendix need improvements. Figure A5-A6, A8-19, difficult to quantify the values. Use more color table or contour interval. The color table is already set to cover the highest difference values. Extending it would only result in lower values to be undistinguishable. The contours are already close to each other, therefore adding more of them would make the plots event harder to read.

7. Why SABER H2O and T are not used in the validation? We thank the reviewer for his/her suggestion to include the comparison with SABER. It could have indeed been done. However, we chose to include in our study only the instruments covering a similar altitude range as our instrument, i.e. the mesosphere and lower thermosphere. Contrary to MIPAS, ACE and MLS, SABER does not measure $H_2O$ in the lower thermosphere (Rong et al., 2019, https://doi.org/10.1016/j.jastp.2019.105099).

Minor comments:

1. Abstract: first line, "temperature is also important tracer" Not really. Temperature is also controlled by solar radiation and cooling. Not a dynamical tracer. Text changed to "Temperature observations are also critical to study middle atmospheric dynamics"

2. page 4, last line, "long wavelength" This has been changed in the updated version of the manuscript.

3. page 6, line 14, what are the physics behind these r0 values? Or just empirically determined? Yes, they are empirically determined. As explained in the text, they are the values that minimize the differences with other instruments. This has now been clarified.

4. page 7, line 9 "and increased methane oxidation...."or sure about it. Any reference? The reference has been changed to Lossow et al.(2017) (see page 1117, Section 4.11, points 4 and 5).

5. page 7, line 14, "geostrophic balance" please double check its accuracy. The accuracy of this sentence has been checked.

6. page 12, line 3, "that is MIPAS", --->"such as" " This has been changed in the updated version of the manuscript.

---

## Author Comment (AC2)

We would like to thank the referees for their constructive comments and suggestions, that helped us improving the quality of our paper. Our detailed replies are included below.

**RC2**

This manuscript discusses newly developed version 3.0 Odin/SMR water vapor and temperature level-2 products. Algorithms used include empirically derived adjustment of the receivers' sideband rejection that improves agreement with correlative measurements. This is subject matter that is appropriate for AMT and that should be entered into the scientific record.

However, the manuscript requires significant revision. Many suggestions have been included in an attached, marked-up pdf.

We are very thankful to the referee for all detailed suggestions included in the supplement pdf. Almost all of them have been applied, with the following exceptions:

- Table 2, page 14: there are no studies published for these v5 datasets. We have calculated the mean vertical resolutions by ourselves. This is why no references have been included in the table.

- Page 2, line 1: We believe the text in the paper is correct. In fact, that is supported by Lossow et al.(2019) (cited in the text) which state that "In the stratosphere and lower mesosphere water vapour has two major sources. One is the transport of water vapour from the troposphere into the stratosphere, for which several pathways exist (Holton et al., 1995; Moyer et al., 1996; Fueglistaler et al., 2009; Sioris et al., 2016). The primary pathway is the slow ascent through the cold tropical tropopause layer, typically accompanied by large horizontal motions."

Moreover, where the comments in the *pdf notes are duplicates of the ones in this document, they have been addressed in the text below.

Correlative datasets from MIPAS, ACE-FTS and MLS should be concisely introduced with appropriate references in a single data section early in the manuscript. Then early discussion in the manuscript to the perceived need to adjust the sideband rejection cold be made less vague. Figures similar to the summary Figures 12 and 13, early in the paper, could make clear the biases in v2.1 that are being addressed with v3.0. A section introducing all the instruments used in this study has been added after the Introduction. In this new section there is a subsection in which we introduce Odin/SMR and one in which we introduce the validation instruments. However, we believe that adding plots showing SMR v2.1 biases early in the text would be superfluous. Those biases can already be seen in Figures 12 and 13 (already cited early in the text), and the suggested plots would be the very similar to 12&13. We therefore chose not to apply this suggestion to avoid the presence of almost-duplicate plots in the paper.

Figures 12 and 13 effectively summarize the content of the "Difference [%]" panels of Figures 8-11 for H2O and A20, A23, A26, and A29 for Temperature in a way that makes comparisons much easier. Similar summary figures could more-concisely convey the content of the correlative-dataset-specific figures, making them

unnecessary. Figures 8-11 and A20, A23, A26, A29 also present the median profiles of H2O concentration and temperature measured by all the instrument. We think it is interesting to show what are the profiles of the actual physical quantities measured by an instrument, other than only showing the differences. Moreover, H2O absolute difference panels provide information on what is the actual VMR difference, which cannot be inferred by the relative difference plots. Finally, those plots also include errors which were not plotted in Figures 12 and 13 for the sake of clarity.

The standard deviation of the median (equation 10) assumes that errors are Gaussian and can be infinitesimally beaten down with more data, making error bars unrealistically small. It would be better to convey some idea of the range of the differences from correlative measurements and some idea of what wiggles in the data are significant. Use of a "bootstrap" method could be useful. For example, are the differences among the three MIPAS datasets significant? The errors plotted in the central and right panels of figures such as Figure 6 are calculated as those of the median of all the single differences between coincident measurements. To be clearer: the absolute difference is calculated as the median of all absolute differences (this is stated in the text, page 13 line 6), and the error is calculated as the error of that standard deviation of the median. A similar calculation is done for the relative difference. From the reviewer's comment, it seems that he/she understood that we are obtaining the plotted errors on the difference profiles from the plotted errors on the concentration/temperature profiles, but that is not the case.

There are several paragraphs associated with individual comparative datasets that describe details of the biases throughout their profiles but that do not provide much insight. These could be reduced/combined in association with plots that combine different correlative data sets. As explained in the reply to an earlier comment (the one suggesting to combine the different comparisons in the same figures), we think that keeping this structure is more accurate, even if it implies a longer descriptive text.

SABER would be a useful additional source of correlative data. The same comment was made by referee #2. Please refer to our answer to his/her comment #7.

There should be discussion of how/why FM13 and FM19 differ. It seems that they put the same H2O spectral line into a spectrometer. Are the spectrometers different? Is this an indication of poorly understood systematics? As already pointed out in the text ("The two FMs use different frontends, that is the set of components denoted by B2 and A1...", page 4 line 28) and in Table 1, the two FMs use the same spectrometer but different frontends (and therefore different optical path, different mixer, etc.)

MLS does not have 1.5 - 3 km vertical resolution; this is rather the resolution of the vertical grid on which data is reported. In the mesosphere, MLS H2O has 3-6 km resolution and temperature has 7-12 km vertical resolution. "Schoeberl et al., 2006" is not the appropriate reference. Cite instead the MLS Data Quality document. We have changed the information about the resolution and the reference as suggested.

Discussion (P6, L11-15) of how new values of $r_0$ were chosen to minimize differences with correlative measurements should be expanded. This section suffers because the correlative datasets have not yet been introduced. It should include figures showing the problem and the improvement. A section introducing the instruments used for validation has been added after the Introduction, as suggested. Figures 12 and 13 show the problem and the improvement, and are referred early in the text.

The r0 values were empirically determined as the ones minimizing the differences between SMR and other instruments, so the problem is shown by the SMR v2.1 bias and the improvement is shown by the SMR v3.0 bias. Both are shown in Figures 12 and 13. This has been clarified in the updated version of the manuscript.

Statements in the conclusions are not all supported by the figures:

P20L3-4: You say MLS and ACE-FTS agree with SMR to -5% -- 0% from 45-80 km, but MLS is -22% at 80 km. ACE-FTS is -10% at 80 km.  Please check thee numbers in the conclusion and abstract. Changed text from: "In particular SMR is in very good agreement with ACE-FTS and MLS in this altitude range" to "In particular SMR is in very good agreement with ACE-FTS and MLS up to 70 km". Furthermore, we have checked that all other statements in the conclusions are consistent with the results shown in the figures.

---

## Author Response (AR2)

Page 1, line 13: "FM13". Please explain what this means or remove, if not necessary in the abstract The sentence has been removed.

Page 2, line 3: I suggest changing "Lyman alpha wavelengths" by "Lyman alpha radiation". Wavelength's can't really photodissociate. Changed as suggested.

Page 2, line 23: "polar mesospheric clouds (Perot et al., 2010)". I don't think this is a good citation for PMCs/NLCs, as there have been many more and much earlier & more general studies on this topic. We kept those citations and added a more general one (Thomas, 2015). Moreover, we now cite those papers indicating them with "e.g".

Page 2, line 29: "H2O" -> "H$_2$O" We changed this to $\mathrm{H_2O}$.

Page 6, line 8: wrong cite command in "found in (Grieco et al., 2020)." Corrected as suggested.

Page 7, caption Fig. 2: "Retrieved concentration profile". Strictly speaking VMR should not be called "concentration", e.g. according to IUPAC. I suggest using the term "mixing ratio" here. This also applies to several other sentences in the paper. Please check. The term "mixing ratio" has now been applied to all the recommended instances.

Page 8, line 14: please add space before "inclination" Corrected as suggested.

Page 10, line 17: Is the summer temperature maximum at the stratopause really driven by GWs? The low polar summer mesopause temperatures yes, but don't think the high temperatures at the stratopause. We changed the sentence from "Summer temperature maxima in the stratopause region and minima in the upper mesosphere result from gravity wave forcing…" to "Summer temperature minima in the upper mesosphere result from gravity wave forcing…"

Page 15, line 8: I suggest mentioning explizitly that "N(z)" is a number and not a number density or mixing ratio. I had to read the sentence several times to get that. We changed the text to " $N(z)$ is the number of differences (absolute or relative) measured at altitude $z$. $\Delta(z)$ is their median."

Page 15, line 12: "outliers data" -> "outliers" Changed as suggested.

Page 20, line 1: "and then decreaseS" Changed as suggested.

Page 22, line 3: "This is to be expected due to the fact that H2O concentration …"

Larger relativ differences are to be expected, but not necessarily large negative differences, right? I suggest changing the statement slightly. We changed the sentence to "These large relative differences are to be expected due to the fact that H2O mixing ratio…"